# A metabolite sensor subunit of the Atg1/ULK complex regulates selective autophagy

A. S. Gross[1,5], R. Ghillebert[1], M. Schuetter[2], E. Reinartz[1], A. Rowland[1], B. C. Bishop[3], M. Stumpe[4], J. Dengjel [4] & M. Graef [1,3] ✉

Cells convert complex metabolic information into stress-adapted autophagy responses. Canonically, multilayered protein kinase networks converge on the conserved Atg1/ULK kinase complex (AKC) to induce non-selective and selective forms of autophagy in response to metabolic changes. Here we show that, upon phosphate starvation, the metabolite sensor Pho81 interacts with the adaptor subunit Atg11 at the AKC via an Atg11/FIP200 interaction motif to modulate pexophagy by virtue of its conserved phospho-metabolite sensing SPX domain. Notably, core AKC components Atg13 and Atg17 are dispensable for phosphate starvation-induced autophagy revealing significant compositional and functional plasticity of the AKC. Our data indicate that, instead of functioning as a selective autophagy receptor, Pho81 compensates for partially inactive Atg13 by promoting Atg11 phosphorylation by Atg1 critical for pexophagy during phosphate starvation. Our work shows Atg11/FIP200 adaptor subunits bind not only selective autophagy receptors but also modulator subunits that convey metabolic information directly to the AKC for autophagy regulation.

Macroautophagy (hereafter autophagy) is a conserved catabolic process critical for cellular homeostasis and health. We are only beginning to understand how cells integrate complex metabolic and functional information to tune stress-adapted autophagy responses. Canonically, cells translate metabolic information into activity patterns of protein kinase networks, which converge on the Atg1/ULK kinase complex (AKC). Assembly and activation of the AKC marks the initial step in hierarchical assembly of highly conserved autophagy-related (Atg) protein machinery resulting in autophagosome biogenesis[1,2]. In budding yeast, the AKC consists of protein kinase Atg1 and scaffold proteins Atg13 and Atg17, in complex with Atg29 and Atg31, and selective autophagy adaptor Atg11 (refs. 3–6). A key step in AKC regulation is inhibitory phosphorylation of Atg13 by target of rapamycin complex 1 (TORC1)[1,7–9]. Upon nutrient starvation, TORC1 inhibition results in Atg13 dephosphorylation driving the formation of supramolecular AKC clusters at sites of autophagosome biogenesis[1,9–11]. In contrast to non-selective autophagy, selective autophagy targets specific substrates including

peroxisomes and mitochondria and initiates autophagosome formation when selective autophagy receptors (SARs) recruit and cluster the AKC on substrate surfaces via binding to Atg11 (FIP200 in mammals)[12–20]. Driven by physical interactions between ubiquitin-like Atg8 proteins, which are covalently attached to autophagic membranes, and substrate-bound receptors, nascent autophagosomes closely encapsulate substrates excluding most other cytosolic components[12,21,22].

Nutrient starvation induces bulk and selective forms of autophagy. How cells coordinate the levels of bulk and selective autophagy in terms of substrates and time in response to diverse nutrient stresses remains an open question. In this Article, we investigated how metabolic signals converge on the AKC and are translated into bulk and selective autophagy in response to phosphate starvation (P-S) or nitrogen starvation (N-S)[23,24]. We discovered differential compositional and functional plasticity of AKC complex in response to the two nutrient stresses. Importantly, we identify phospho-metabolite

[1]Max Planck Research Group of Autophagy and Cellular Ageing, Max Planck Institute for Biology of Ageing, Cologne, Germany. [2]Max Planck Research Metabolomics Core Facility, Max Planck Institute for Biology of Ageing, Cologne, Germany. [3]Department of Molecular Biology and Genetics, Cornell University, Ithaca, NY, USA. [4]Department of Biology, University of Fribourg, Fribourg, Switzerland. [5]Present address: Gregor Mendel Institute of Molecular Plant Biology, Vienna Biocenter, Vienna, Austria. ✉e-mail: Martin.Graef@cornell.edu

sensor Pho81 as a modulator of selective autophagy, which directly conveys metabolic information to the AKC to regulate the turnover of peroxisomes during P-S.

## Results

### Metabolite sensor Pho81 binds the AKC

We asked whether induction of autophagy has similar compositional and functional requirements for the AKC in response to P-S or N-S. To address this question, we compared autophagic flux in wild-type (WT) cells with cells deficient for kinase Atg1, selective autophagy adaptor Atg11 or scaffold proteins Atg13 or Atg17, Atg29 and Atg31 (Fig. 1a). We monitored autophagy flux by turnover of 2GFP–Atg8, which is covalently attached to autophagic membranes and cleaved within vacuoles to generate free green fluorescent protein (GFP)[25]. The absence of Atg1 prevented any autophagic turnover in P-S or N-S (Fig. 1b). Interestingly, we detected substantial autophagy flux in *Δatg11* cells during P-S, in contrast to previous work[26]. Strikingly, while essential for N-S-induced autophagy consistent with previous data[1,3], *Δatg13* and *Δatg17* cells showed significant turnover of 2GFP–Atg8 during P-S (Fig. 1b). Cells depended on Atg11 in the absence of Atg13 or Atg17, because *Δatg11Δatg13* and *Δatg11Δatg17* cells were completely inhibited for autophagy during P-S (Fig. 1b). Strikingly, *Δatg13Δatg17Δatg29Δatg31* cells maintained significant 2GFP–Atg8 turnover compared with WT cells after P-S (24 h) (Fig. 1b), suggesting the existence of a minimal Atg1–Atg11 complex sufficient to support autophagic activity. Taken together, we find a strikingly differential functional and compositional plasticity of the AKC in response to P-S and N-S.

Since cells displayed significant autophagy in absence of Atg13 or the whole Atg13–17–29–31 subcomplex during P-S, we asked whether additional as yet unidentified components may compensate for their function at the AKC. To address this question, we determined the protein interactome of C-terminally triple-GFP-tagged variants of Atg1, of Atg1 in the absence of Atg11, and of Atg11 by performing co-immunopurifications coupled with mass spectrometry (CoIP–MS) after P-S (8 h). Interestingly, in addition to Atg8 and canonical AKC components, Atg1–3GFP showed significant enrichment for metabolite sensor Pho81 (Fig. 1c). Pho81 functions as a phospho-metabolite sensor and inhibitor of cyclin-dependent Pho80–Pho85 kinase[26–29]. Notably, Pho81 interaction was lost for Atg1–3GFP isolated from *Δatg11* cells (Fig. 1c), demonstrating Atg11-dependent AKC binding of Pho81. Consistently, we found strong enrichment of Pho81 and all canonical AKC subunits when we profiled the protein interactome of Atg11–3GFP (Fig. 1c). Taken together, these data reveal an Atg11-dependent physical AKC interaction of Pho81 during P-S.

To test whether Pho81 interacts with the AKC in a P-S-dependent manner, we performed CoIP of cells co-expressing plasmid-encoded GFP–*ATG11* and *PHO81*–mCherry under growing conditions or after P-S (24 h). We detected a basal interaction of Pho81–mCherry with GFP–Atg11, but not with GFP alone (Fig. 1d). The interaction increased tenfold in the absence of phosphate, indicating P-S-induced binding between Pho81 and Atg11 (Fig. 1d). To test whether Pho81 interacts with Atg11 at sites of autophagosome formation, we co-expressed GFP–*ATG11* with *PHO81*–*mCherry* under its endogenous (Fig. 1e) or overexpression promoter (*ADH1*) (Fig. 1f). Pho81–mCherry formed peripheral structures after P-S (4 h), which co-localized with GFP–Atg11 (Fig. 1e,f). In strains co-expressing *2GFP*–*ATG8* and plasmid encoded *PHO81*–*mCherry* under an *ADH1* overexpression promoter, we also observed GFP–Atg8-positive Pho81–mCherry structures after P-S (4 h) (Extended Data Fig. 1a). Taken together, our data show that Pho81 physically interacts with the AKC in an Atg11-dependent manner and localizes to Atg8- and Atg11-marked sites of autophagosome formation during P-S.

### Pho81 promotes pexophagy

We next aimed at defining the function of Pho81 at the AKC for autophagy. The functions and substrates of autophagy during P-S have been unknown. To identify potential substrates of autophagy in an unbiased manner, we characterized the autophagy-dependent changes in the yeast proteome during P-S. We performed high-resolution quantitative whole cell proteomics of WT, *Δatg1*, *Δatg11*, *Δpho81* and *Δpho81Δatg11* cells after P-S (24 h). We identified 3,500–4,500 proteins in a quantitative manner corresponding to >70–90% of the expressed yeast proteome (Fig. 2a,b). During P-S, *Δatg1* and *Δatg11* cells significantly accumulated 263 and 294 proteins, respectively (Fig. 2a). Among significantly enriched proteins, we identified 136 and 209 mitochondrial and 23 and 28 peroxisomal proteins for *Δatg1* and *Δatg11* cells compared with WT cells, respectively (Fig. 2a). These data indicate a role for selective autophagy in mitochondria and peroxisome turnover during P-S. Peroxisomal and mitochondrial proteins also accumulated in *Δpho81* and *Δpho81Δatg11* cells compared with WT cells (Fig. 2b), raising the possibility that Pho81 plays a role in selective autophagy of mitochondria and peroxisomes during P-S.

We monitored autophagic degradation of mitochondrial or peroxisomal membrane proteins Om45–GFP or Pex11–GFP, respectively, in comparison with 2GFP–Atg8 turnover. Pho81 did not affect autophagic membrane turnover measured by 2GFP–Atg8 consistent with previous work (Fig. 2c)[24]. However, in line with our proteomics data, both, mitochondria and peroxisomes were significantly degraded by autophagy (Fig. 2c). Importantly, while *PHO81* deletion led to a mild but significant reduction in Om45–GFP turnover, it robustly impaired Pex11–GFP degradation after P-S (Fig. 2c), indicating Pho81 functions in mitophagy and, more critically, in pexophagy. The selective adaptor Atg11, pexophagy-receptor Atg36 and co-receptor Pex3 are essential for pexophagy during N-S[30,31]. We found that pexophagy strictly depended on the presence of all three factors during P-S (Fig. 2d,e), indicating that cells use the same core pexophagy machinery during nitrogen and P-S. In line with autophagic turnover, the number of peroxisomes declined in WT cells after P-S (24 h) compared with non-starved cells (Fig. 2f). In contrast, reduced pexophagy in *Δpho81*, *Δatg11* and *Δpho81Δatg11* cells correlated with significant accumulation of peroxisomes compared with WT cells after P-S (24 h) (Fig. 2f). Taken together, these data identify peroxisomes as selective cargo for P-S-induced autophagy and show that Pho81 significantly promotes pexophagy mediated by canonical Atg11, Atg36 and Pex3 pexophagy machinery to regulate the abundance of peroxisomes.

### Pho81 does not function as a canonical autophagy receptor

The physical interaction of Pho81 with Atg11 and its association with sites of autophagosome formation suggest a direct role for Pho81 in promoting pexophagy during P-S. We tested whether Pho81 may function as a SAR. To assess whether Pho81 spatially associates with peroxisomes, we performed fluorescence imaging of endogenous Pho81–GFP and peroxisomes marked by BFP–eSKL[32]. Interestingly, while we did not detect Pho81 directly on peroxisomes, peripheral peroxisomes often localized in close proximity to cortical Pho81 structures upon P-S independent of the presence of Atg11 or Atg36 (Fig. 2g). These data place Pho81 in spatial proximity to peroxisomes consistent with a role in pexophagy, but suggest that Pho81 does not mark peroxisome surfaces for degradation in contrast to obligate pexophagy-receptor Atg36 or co-receptor Pex3 (refs. 30,31).

SARs bind both, Atg11 to cluster the AKC and Atg8 to tether selective cargos to growing phagophores[20]. Yeast-two-hybrid (Y2H) analysis confirmed the physical interaction of activator domain (AD)-Pho81 with binding domain (BD)-Atg11 independent of Atg36 (Fig. 3a and Extended Data Fig. 1b). Interestingly, we did not detect an AD-Pho81 and BD-Atg8 interaction (Fig. 3a). Notably, consistent with previous data[33], pexophagy-receptor AD-Atg36 interacted with both, BD-Atg11 and BD-Atg8 (Fig. 3a). These data suggest Pho81 positively regulates Atg11–Atg36-driven pexophagy in a manner functionally distinct from the mechanisms of canonical SARs.

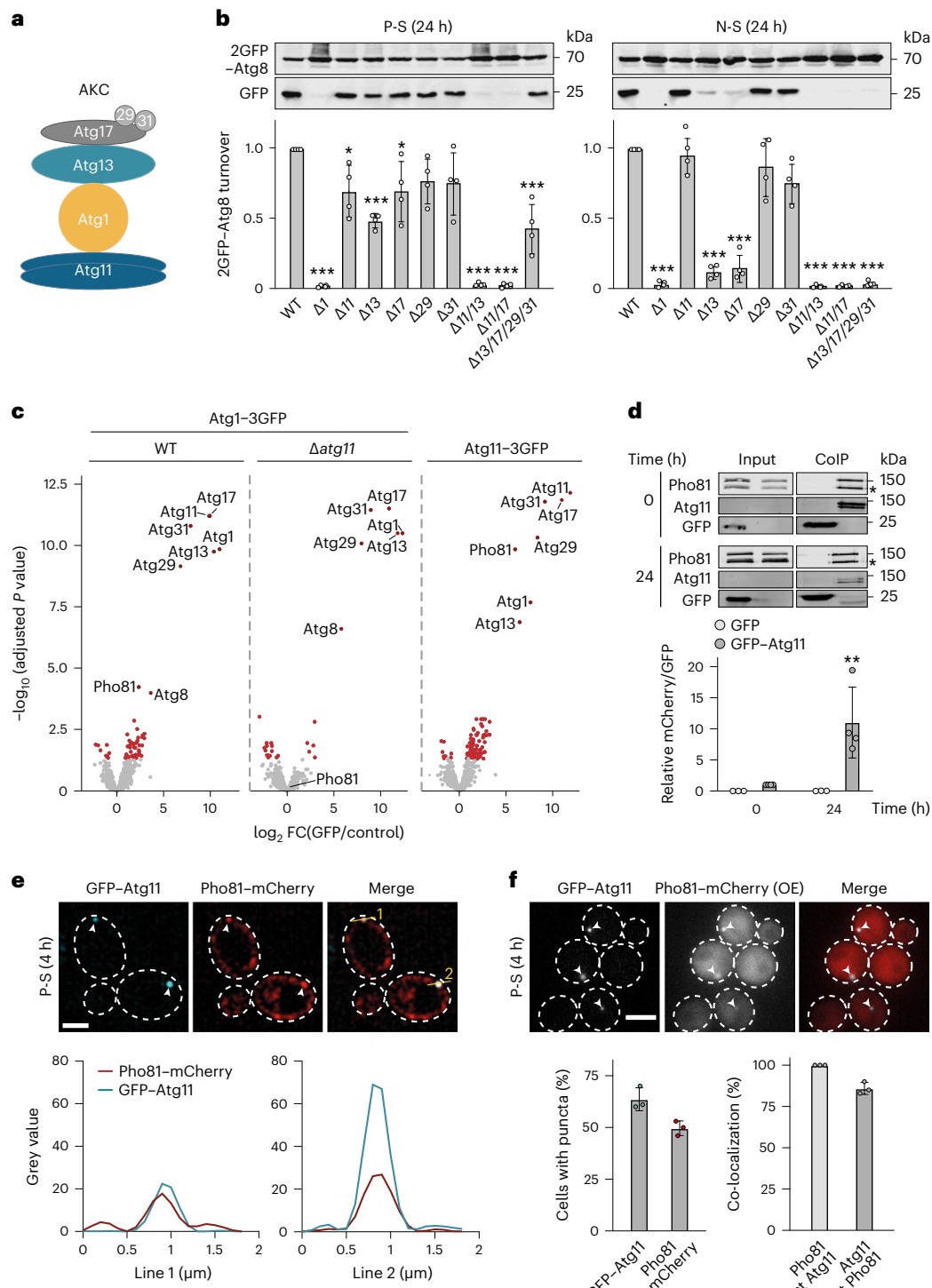

**Fig. 1 | Pho81 binds to the AKC via Atg11 during P-S. a**, Schematics of the AKC. **b**, 2GFP–Atg8 turnover in indicated strains after P-S or N-S (24 h). Cells were analysed by whole cell extraction and western blot analysis using an α-GFP antibody. Data are mean ± s.d. (*n* = 4 biologically independent experiments). **c**, Fold changes (FC) in protein abundance relative to negative control after CoIP-MS of indicated 3GFP-tagged proteins after P-S (8 h). Significantly enriched proteins are shown in red (*n* = 4 biologically independent experiments). **d**, CoIP after GFP pulldown from cells harbouring plasmid-encoded *GFP–ATG11* and *PHO81–mCherry* during growth (0 h) or P-S (24 h). Quantifications show relative levels of Pho81–mCherry over GFP and GFP–Atg11 normalized to 0 h Pho81–mCherry/GFP–Atg11. Asterisk indicates an unspecific band. Samples were derived from the same experiment and processed in parallel. Data are mean ± s.d. (*n* = 4 biologically independent experiments). **e**, Fluorescence imaging of cells expressing plasmid encoded *GFP–ATG11* (cyan) and *PHO81–mCherry* (red) after P-S (4 h). Intensity plots along

yellow lines show grey values for indicated proteins. Data are representative of three biologically independent experiments. Scale bar, 5 μm. **f**, Fluorescence imaging of cells expressing plasmid encoded *PHO81–mCherry* under *ADH1* promoter (OE) and *GFP–ATG11* under its endogenous promoter after P-S (4 h). Scale bar, 5 μm. Quantification of Pho81–mCherry and GFP–Atg11 puncta per cell (left) and the respective co-localization (right). Data are mean ± s.d. (*n* = 100 cells examined over three independent experiments). Statistical significance was assessed using one-way ANOVA followed by Tukey's multiple comparisons test. *P* values relative to WT: Δ*atg11* *P* = 0.0165; Δ*atg17* *P* = 0.0203; ****P* < 0.0001, except Δatg13Δatg17Δatg29Δatg31 *P* = 0.0002 (**b**), two-sided statistical testing with adjustment for multiple testing within a comparison performed using limma[49,50] (**c**) and two-way ANOVA followed by Tukey's multiple comparisons test (**d**). *P* values relative to 0 h Pho81–mCherry/GFP–Atg11: ***P* = 0.0049. Source numerical data and unprocessed blots are available in Source data.

## Pho81 functionally interacts with Atg13 during pexophagy

We asked whether Pho81 regulates the AKC as a modulator subunit to promote pexophagy. We observed that, in contrast to N-S, autophagic membrane turnover did not strictly depend on the presence of Atg13 or Atg17 during P-S (Fig. 1b). We examined whether Pho81 functionally interacts with Atg13 and Atg17 during pexophagy. Pex11–GFP turnover was fully inhibited in Δatg1 and Δatg11 cells (Fig. 3b), demonstrating that pexophagy requires the Atg1 kinase and Atg11 adaptor during P-S. However, pexophagy was only partially reduced in Δatg13, Δatg17 and Δpho81 cells compared with WT cells (Fig. 3b, c). Importantly, Δpho81Δatg17 and Δpho81Δatg13 cells showed strongly impaired pexophagy upon P-S (Fig. 3c), indicating functional interactions between Pho81 and Atg13 or Atg17. In contrast, during N-S, Δpho81 cells were only mildly affected, while Atg13 was required for pexophagy (Fig. 3d), demonstrating differential compositional AKC requirements for pexophagy upon different nutrient stresses. Turnover of autophagic membranes (2GFP–Atg8) or bulk autophagy (cytosolic tandem GFP; 2GFP) depended on the presence of Atg13 without additional contribution by Pho81 during P-S (Fig. 3e,f). Taken together, these data uncover a functional interaction of Pho81 and Atg13 specific for pexophagy during P-S. Analysing the cytological relationship, we observed co-localization of Pho81–GFP and Atg13–mCherry in phosphate-starved cells (Fig. 3g). Interestingly, the absence of Atg13 significantly increased the localization of Pho81–GFP to Atg1–mCherry-marked puncta (Fig. 3h), suggesting competitive recruitment of Pho81 and Atg13 to sites of autophagosome formation.

To define the regulatory mechanisms underlying the differential requirements for Atg13 and Pho81 for pexophagy during nitrogen and P-S, we performed unbiased whole cell phosphoproteomics in WT, Δpho81, Δatg13 and Δatg13Δpho81 cells. We identified ~2,800 significantly differentially changed phosphosites between P-S and N-S (Fig. 4a,b), indicating divergent regulatory cell states. To begin to dissect regulatory differences, we focused on representative known phosphorylation sites of TORC1 and Atg1 (ref. 7). Notably, independent of Pho81 or Atg13, TORC1 was strongly inhibited upon N-S, but retained significant activity during P-S (Fig. 4c). Consistently, the analysis of phosphorylation sites of Atg1 confirmed Atg13-dependent activation of Atg1 kinase during N-S independent of Pho81 (Fig. 4c). However, P-S induced a differential phosphorylation pattern of Atg1 sites in dependence of both, Atg13 and Pho81 (Fig. 4c), suggesting complex regulatory differences of autophagy in response to both nutrient stresses.

Based on the differential activity of TORC1 during nitrogen and P-S, we examined AKC regulation in dependence of TORC1-regulated Atg13 phosphorylation[1,9–11]. Interestingly, global phosphorylation of Atg13 was largely maintained in WT and Δpho81 cells upon P-S consistent with previous data (Fig. 4d)[24]. In contrast, TORC1 inhibition upon N-S led to rapid dephosphorylation of Atg13 in WT and Δpho81 cells, as described previously (Fig. 4d)[1,7]. TORC1 inhibition by rapamycin treatment resulted in Atg13 dephosphorylation during P-S similar to

N-S, independent of Pho81 (Fig. 4d). Thus, consistent with our phosphoproteome analysis, these data confirm that TORC1 retains significant activity during P-S (Fig. 4d). We hypothesized that phosphorylated Atg13 is only partially activated and, as a consequence, cells depend on Pho81 to induce pexophagy during P-S. To test this notion, we treated cells with rapamycin to inhibit TORC1 and fully activate Atg13 during P-S[1]. Strikingly, rapamycin-treated Δpho81 cells displayed WT-like pexophagy during P-S (Fig. 4e), indicating that full Atg13 activation renders pexophagy independent of Pho81. Pexophagy still occurred in an Atg11-dependent manner during P-S upon TORC1 inhibition, excluding rapamycin-induced non-selective (bulk) pexophagy. The effect of rapamycin exclusively depended on the presence of Atg13 (Fig. 4e), indicating that TORC1 inhibition did not affect Pho81-mediated pexophagy. In turn, increased TORC1 activity in the absence of the TORC1-inhibitor kinase Npr2 made pexophagy largely Pho81 dependent (Fig. 4f). These data indicate that TORC1-dependent phosphorylation of Atg13 determines the dependence of pexophagy on Pho81. Our data support a model in which TORC1 retains significant activity during P-S, resulting in a high level of Atg13 phosphorylation. To compensate for only partially activated Atg13, the induction of pexophagy requires the recruitment and function of Pho81 at the AKC. Thus, the differential compositional and functional plasticity of the AKC we observed during P-S and N-S is at least in part based on the conditional function of the modulator subunit Pho81.

## Pho81 regulates Atg11 phosphorylation critical for pexophagy

We aimed at determining the molecular nature of the physical interaction of Pho81 and Atg11. Pho81 is composed of an N-terminal SPX domain, a minimal domain (MD), an ankyrin domain and a C-terminal glycerophosphodiester-phosphodiesterase (GP-PDE) domain (Fig. 5a)[26,28]. Ankyrin domains consist of helix-turn-helix-loop repeats and commonly form a platform for protein–protein interactions defined by highly versatile loop sequences[34]. We interrogated the potential role of the Pho81 ankyrin domain for Atg11 interaction using Y2H. We created Pho81 variants by replacing either the entire ankyrin domain (Pho81[A]) or exchanging individual loops (Pho81[LA1] to [LA4]) with the corresponding sequences of the ankyrin domain of Akr1 (Fig. 5a). The exchange of the whole ankyrin domain (Pho81[A]) impaired Pho81–Atg11 interaction (Fig. 5b). Changes in the loop of the first Pho81 ankyrin repeat completely prevented Pho81–Atg11 interaction, but not in loops of ankyrin repeat 2–4. These data reveal a critical role for the first loop of the ankyrin domain of Pho81 for Atg11 binding. Next, we co-expressed plasmid-encoded PHO81–mCherry under the ADH1 promoter with GFP–ATG11 under its endogenous promoter and monitored co-localization during growing conditions using fluorescence microscopy. We observed quantitative recruitment of Pho81–mCherry to GFP–Atg11 puncta, establishing Pho81–mCherry puncta formation as a proxy for Atg11 interaction (Fig. 1f). Consistent with our Y2H data, puncta formation specifically required Pho81 ankyrin repeat

---

**Fig. 2 | Pho81 positively regulates pexophagy during P-S. a,b,** Whole cell TMT proteomics analysis of indicated strains compared with WT after P-S (24 h) (n = 3 biologically independent experiments): significant proteome changes in Δatg1 and Δatg11 cells compared with WT, where volcano plots show significant fold changes (FC) above horizontal dotted line and top 20 proteins with the highest adjusted P value are labelled (**a**); significant proteome changes in Δpho81, Δatg11 and Δatg11Δpho81 cells compared with WT (**b**). **c,** 2GFP–Atg8, Om45–GFP or Pex11–GFP turnover in WT and Δpho81 cells upon P-S (24 h). Data are mean ± s.d. (n = 6 biologically independent experiments). **d,e,** Pex11–GFP turnover in indicated strains after P-S (24 h). Data are mean ± s.d. (**d**, n = 11; **e**, n = 3 biologically independent experiments). **f,** Fluorescence imaging of indicated strains expressing PEX11–GFP after P-S (0 and 24 h). Images represent maximum intensity Z-stack projections. Right: quantification of peroxisomes per cell. Box-and-whiskers plots: the boxes extend from the 25th to the 75th percentile spanning the IQR, whiskers show minimal and maximal values, black line in

the middle of the boxes represents the median (n = 100 cells examined over four independent experiments). **g,** Fluorescence imaging of indicated strains expressing genomic PHO81–GFP and pRS315–BFPeSKL to visualize peroxisomes during growth and after P-S (4 h). Arrowheads indicate Pho81–GFP positive (white) or negative (purple) peroxisomes. Data are representative of three biologically independent experiments. Scale bars, 5 μm (**f** and **g**). Statistical significance was assessed using two-sided statistical testing with adjustment for multiple testing within a comparison performed using limma[49] (**a** and **b**), two-tailed t-test **P = 0.0005 (**c**), one-way ANOVA followed by Tukey's multiple comparisons test, P values relative to WT: **P = 0.0033; ***P < 0.001 (**d** and **e**), and two-way ANOVA followed by Šídák's multiple comparisons test, P values relative to WT: ***P < 0.0001 (24 h); P values 24 versus 0 h timepoints: #, P = 0.0319; ##, P = 0.0014; ###, P < 0.0001 (**f**). Source numerical data and unprocessed blots are available in Source data.

loop 1, but not loop 2 and loop 3 (Extended Data Fig. 1c). The Pho81[LA4] variant allowed Atg11 interaction in Y2H, but failed to form foci probably due to its nuclear localization (Fig. 5b and Extended Data Fig. 1c).

In summary, these data demonstrate that Pho81 binding and recruitment to Atg11-marked sites of autophagosome biogenesis specifically depend on the loop of the first ankyrin repeat of Pho81.

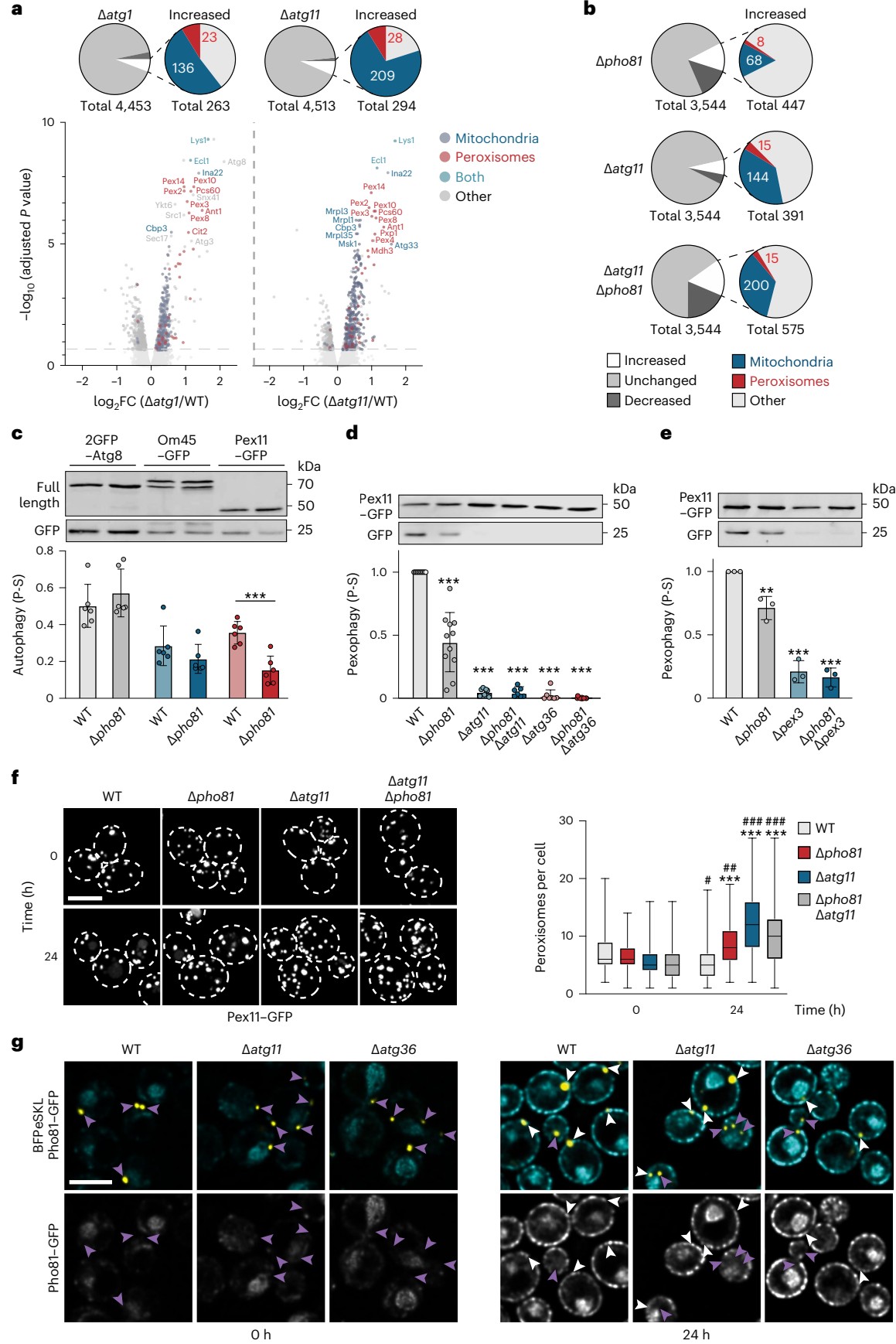

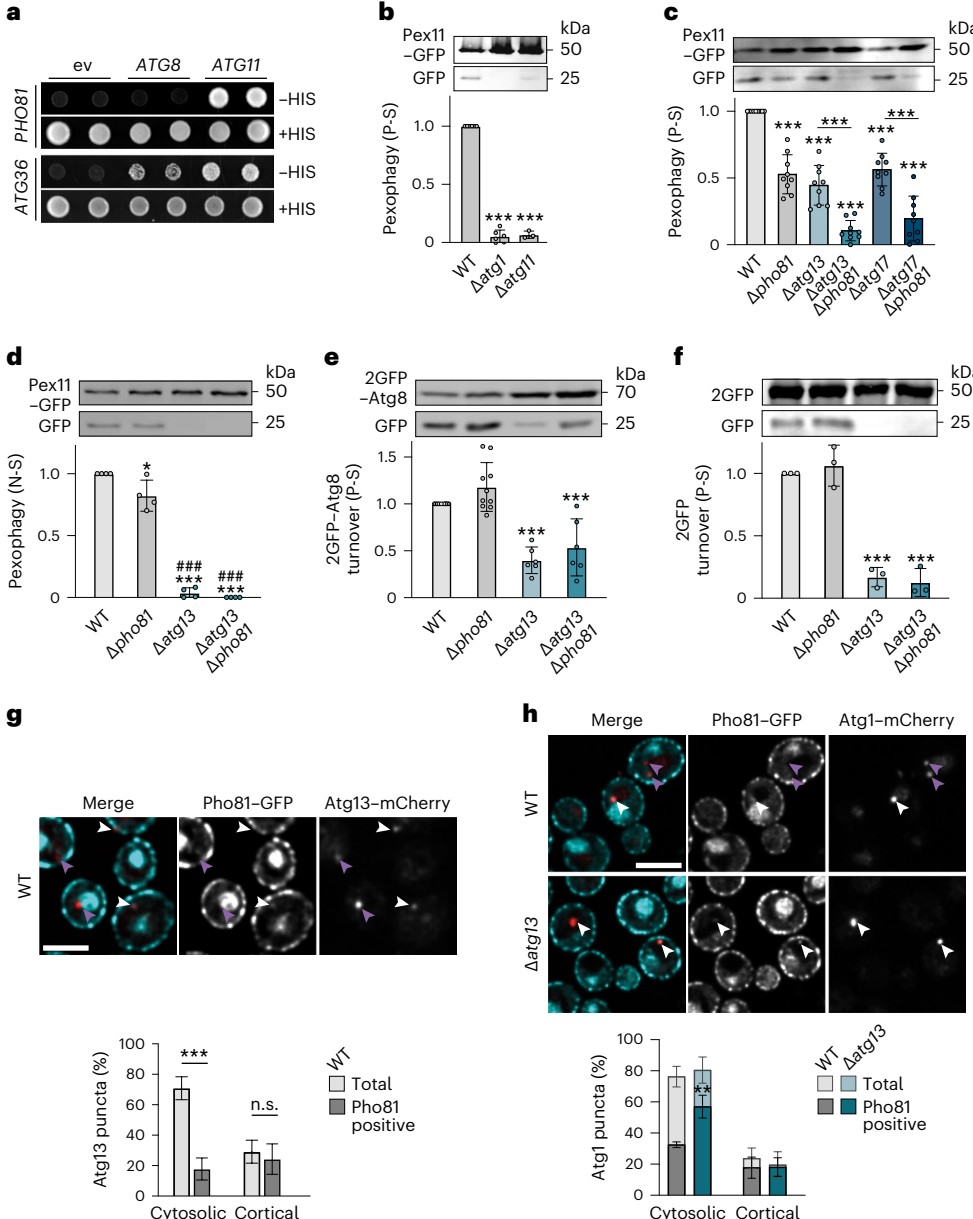

**Fig. 3 | Pho81 functionally interacts with Atg13 and Atg17 during pexophagy upon P-S. a**, Y2H assay of cells expressing pGAD-*PHO81* or pGAD-*ATG36* in combination with pGBDU (empty vector, ev), pGBDU-*ATG8* or pGBDU-*ATG11* grown on SD + HIS or SD − HIS medium. **b–d**, Pex11–GFP turnover in indicated strains after P-S (24 h) (**b**, *n* = 3–6; **c**, *n* = 9 biologically independent experiments) or after N-S (24 h) (*n* = 4 biologically independent experiments) (**d**). Data are mean ± s.d. **e**, 2GFP–Atg8 turnover in indicated strains after P-S (24 h). Data are mean ± s.d. (*n* = 6 biologically independent experiments). **f**, 2GFP turnover in indicated strains after P-S (24 h). Data are mean ± s.d. (*n* = 3 biologically independent experiments). **g**, Fluorescence imaging of WT cells expressing genomic *PHO81*–GFP and *ATG13*–mCherry after P-S (4 h). Arrowheads indicate Pho81–GFP positive (white) or negative (purple) Atg13 foci. Scale bar, 5 μm. Quantification of Atg13–mCherry and Pho81–GFP co-localization in WT

cells. Data are mean ± s.d. (*n* = 150 cells examined over three independent experiments). **h**, Fluorescence imaging of indicated strains expressing genomic *PHO81*–GFP and *ATG1*–mCherry after P-S (4 h). Arrowheads indicate Pho81–GFP-positive (white) or Pho81–GFP-negative (purple) Atg1 puncta. Scale bar, 5 μm. Quantification of Atg1–mCherry and Pho81–GFP co-localization in WT and Δ*atg13* cells. Data are mean ± s.d. (*n* = 300 cells examined over three independent experiments). Statistical significance was assessed using one-way ANOVA followed by Tukey's multiple comparisons test. *P* values relative to WT or as indicated: \*\*\**P* < 0.0001 except \**P* = 0.0494 (**d**) and *P* values relative to Δ*pho81* ###, *P* < 0.0001 (**b–f**). Two-way ANOVA followed by Tukey's multiple comparisons test. *P* values relative to WT or as indicated: \*\*\**P* = 0.0002 (**g**), \*\**P* = 0.0078 (**h**), non-significant (n.s.). Source numerical data and unprocessed blots are available in Source data.

Alignment of known Atg11- and FIP200-binding regions (FIRs) revealed a cluster of negatively charged aspartic and glutamic acid (D or E) or phosphorylatable serine or threonine (S or T) residues proximal to two hydrophobic amino acids (P, V, L, I or F) (Fig. 5c), consistent with previous data[18,35–37]. Interestingly, we identified potential FIRs in the loop of the first Pho81 ankyrin repeat (amino acids 451–461) and at amino acids 806–816 located between the

minimal and GP-PDE domain (Fig. 5c). We generated Pho81^DD (D453K/D456K), Pho81^T (T461A), Pho81^DDT (D453K/D456K/T461A) and Pho81^DE (D806K/E808K) variants (Fig. 5d). Y2H analysis revealed a strict requirement for D453/D456 in loop 1 (Pho81^DD) for the Pho81–Atg11 interaction. Additionally, we found a partial reduction in Atg11 binding for the Pho81^DE variant (Fig. 5e). Residue T461 in Pho81 is a confirmed phosphorylation site at a similar position as S119 site in the FIR motif of

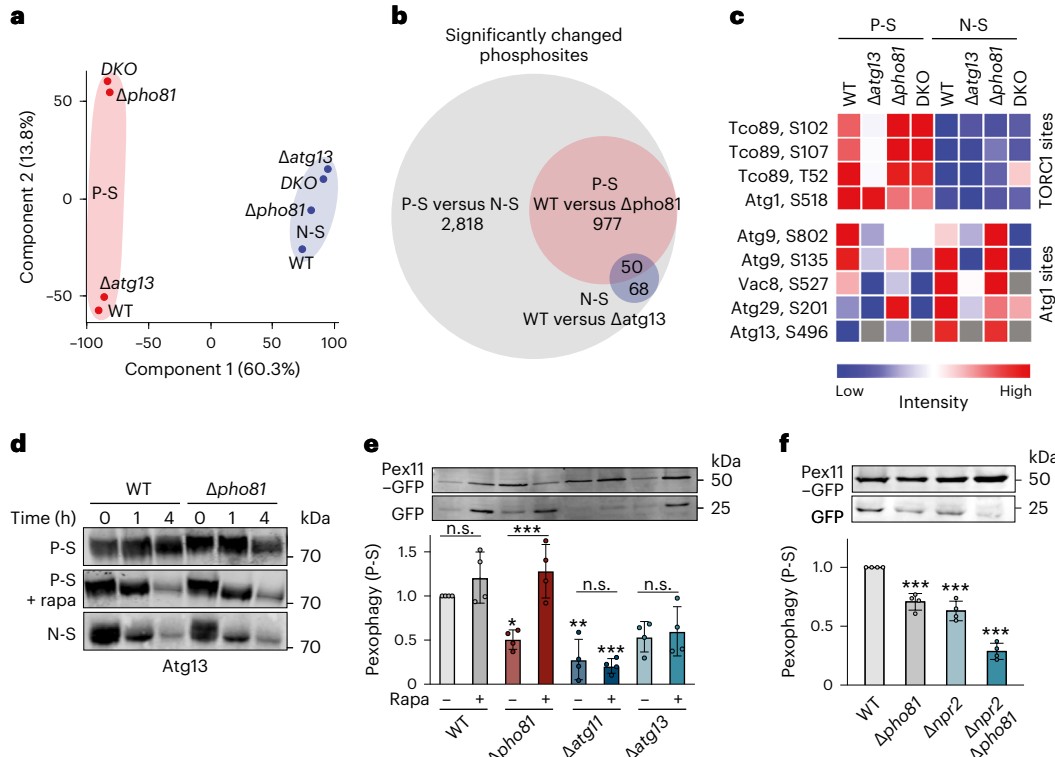

**Fig. 4 | Atg13 phosphorylation by TORC1 determines dependence of pexophagy on Pho81 during P-S. a**, Principal component analysis of phospho-proteomics data for WT and indicated cells upon P-S (red) and N-S (blue) after 8 h. **b**, Significantly changed phosphosites during P-S and N-S (grey), WT and *Δpho81* after P-S (red), or WT and *Δatg13* after N-S (blue) after 8 h. **c**, Phosphorylation of representative TORC1 kinase and Atg1 kinase sites in indicated cells upon P-S and N-S (8 h). Data in **a**–**c** are derived from four biologically independent experiments. **d**, Phosphorylation of endogenous Atg13 in WT and *Δpho81* cells during P-S ± rapamycin or N-S at indicated timepoints. Samples were analysed by whole cell extraction and western blot using an α-Atg13 antibody (*n* = 3 biologically

independent experiments). **e**, Pex11–GFP turnover in indicated strains after P-S (24 h) ± rapamycin. Data are mean ± s.d. (*n* = 4 biologically independent experiments). **f**, Pex11–GFP turnover in indicated strains after P-S (24 h). Data are mean ± s.d. (*n* = 4 biologically independent experiments). Statistical significance was assessed using two-way ANOVA followed by Tukey's multiple comparisons test. *P* values relative to WT-rapa or as indicated *$P$ = 0.0494, **$P$ = 0.0014, Δpho81 − versus + rapa ***$P$ = 0.0006, Δatg11 + rapa versus WT − rapa ***$P$ = 0.0004 (**e**) and one-way ANOVA followed by Tukey's multiple comparisons test, *P* values relative to WT, ***$P$ ≤ 0.0002 (**f**), non-significant (n.s.). Source numerical data and unprocessed blots are available in Source data.

mitophagy receptor Atg32 supporting Atg11 interaction[33,38–40]. Pho81[T] displayed a partial Atg11 binding defect in Y2H (Fig. 5d,e), suggesting T461 and D806/E808 sites are not essential but support Pho81–Atg11 interaction (Fig. 5d,e). CoIP–MS of plasmid-encoded *GFP–ATG11* from cells expressing either *PHO81*- or *pho81[DD]–mCherry* validated D453/D456-dependent Pho81–Atg11 interaction after P-S (8 h) (Fig. 5f). Notably, interactions of core AKC components with GFP–Atg11 were unchanged in the presence of Pho81 or Pho81[DD], demonstrating Pho81–Atg11 interaction-independent AKC assembly (Fig. 5f). Thus, our data identify a FIR within the first repeat of the ankyrin domain of Pho81 required for Atg11-dependent binding to the AKC.

Identification of an Atg11-binding-deficient Pho81 variant allowed us to specifically assess the functional relevance of the Pho81–AKC interaction for pexophagy. Expression of plasmid-encoded *PHO81* under its endogenous promoter restored pexophagy in *Δpho81* cells, while *pho81[DD]* and *pho81[DDT]* did not (Fig. 5g). Expression of *pho81[T]* did not significantly affect pexophagy, suggesting post-translational modifications of Pho81 at T461 are not critical for Pho81-dependent pexophagy (Fig. 5g)[33,37,40]. Notably, impaired Pho81–Atg11 interaction through mutation of D453/D456 strongly blocked pexophagy in absence of Atg13 (Fig. 5h), indicating Pho81 binding to Atg11 is required for its compensatory function for pexophagy. To test whether effects of the Pho81 variants on pexophagy were caused by potential changes in the PHO pathway, we analysed pexophagy in the absence of the transcription factor *PHO4*, essential for PHO pathway activation[27,41,42]. In contrast to cells expressing *PHO81*, the *pho81[DD]* variant failed to restore

pexophagy in *Δpho81Δpho4* cells (Extended Data Fig. 1f). Thus, independent of its role in the PHO pathway, Pho81 functions as a modulator subunit of the AKC without affecting core subunit assembly.

To define the molecular mechanisms of how Pho81 regulates the AKC for pexophagy, we analysed changes in phosphosites of isolated AKC subunits after CoIP–MS of GFP–Atg11 in dependence of AKC-bound Pho81 during P-S. Strikingly, phosphorylation of S121, S613 and S935 within Atg11 was significantly reduced in the presence of Atg11-binding-deficient Pho81[DD] compared with Pho81 (Fig. 5i). S613 and S935 have previously been identified as Atg1 kinase phosphosites[7], indicating Pho81–Atg11 interaction promotes Atg11 phosphorylation by Atg1 kinase. To assess the functional relevance of the three phosphosites, we expressed plasmid-encoded *ATG11* or *atg11[3SA]*, carrying non-phosphorylatable S121A, S613A, S935A exchanges, in *Δatg11* or in *Δatg11Δpho81* cells. Strikingly, *atg11[3SA]* expression almost completely blocked Pho81-dependent pexophagy compared with *ATG11* in an epistatic manner with *Δpho81* (Fig. 5j). Taken together, our data support a model in which Pho81 functions as a modulator subunit of the AKC by binding to Atg11 to promote Atg11 phosphorylation by Atg1 kinase critical for Atg11–Atg36-mediated pexophagy during P-S.

## Metabolite-sensing SPX domain of Pho81 promotes pexophagy

SPX domains are conserved sensors of phospho-metabolites from yeast to human[27,28,43]. They form small (135–380 amino acids) N-terminal

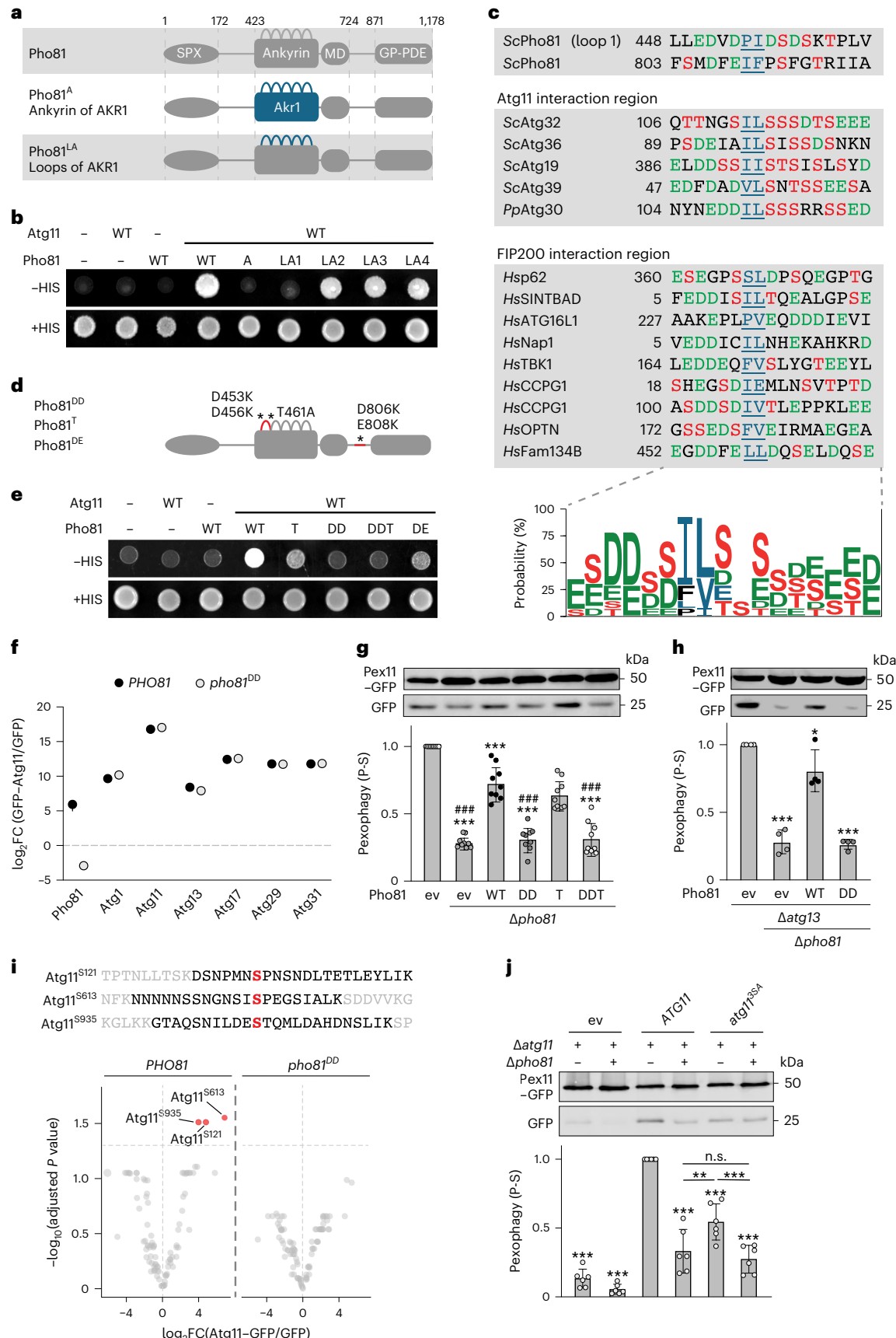

structures with two long core helices, which interact with inositol polyphosphates ($InsP_6$) and pyrophosphates ($InsP_{7/8}$) via a conserved cluster of amino acid residues (Fig. 6a,b)[28,42,44]. We hypothesized that the SPX domain of Pho81 directly transmits phospho-metabolic information to the AKC to regulate pexophagy during P-S. We generated a Pho81$^{\Delta SPX}$ variant without SPX domain (1–172 amino acids) and

**Fig. 5 | Pho81 interacts with Atg11 via an Atg11/FIP200 interaction motif to promote phosphorylation of Atg11 required for pexophagy. a**, Schematics of Pho81 and ankyrin repeat variants; Pho81$^A$ (Pho81 ankyrin repeat replaced by Ankyrin repeat of Akr1); Pho81$^{LA1}$ to Pho81$^{LA4}$ (Pho81 ankyrin repeat loops 1 to 4 replaced by corresponding loop of Akr1 Ankyrin repeat loops 1 to 4).
**b**, Y2H of cells expressing indicated pGAD-*PHO81* variants and pGBDU-empty vector (−) or *ATG11* were grown on SD + HIS or SD − HIS medium. **c**, Consensus motif of Atg11/FIP200 interaction region with homology logo (WebLogo 3.0)[50]. based on alignment of indicated Atg11/FIP200 interaction regions and corresponding amino acid probability. *Sc*, *Saccharomyces cerevisiae*; *Pp*, *Pichia pastoris*; *Hs*, *Homo sapiens*. **d**, Schematics of Pho81$^{DD}$, Pho81$^T$ and Pho81$^{DE}$.
**e**, Y2H of cells expressing indicated pGAD-*PHO81* variants and pGBDU-empty vector (−) or *ATG11* were grown on SD + HIS or SD − HIS medium. **f**, Enrichment of AKC subunits after GFP–Atg11-based CoIP–MS from Δ*pho81* cells expressing plasmid-encoded *PHO81*–mCherry or *pho81$^{DD}$*–mCherry. Control is GFP.

Data are mean ± s.d. (*n* = 3–4 biologically independent experiments). Statistical analysis is presented in Supplementary Table 2. FC, fold change. **g,h**, Pex11–GFP turnover in indicated strains harboring empty vector (ev) or expressing plasmid-encoded *PHO81* variants after P-S (24 h). Data are mean ± s.d. (**g**, *n* = 9; **h**, *n* = 4 biologically independent experiments). **i**, Atg11 peptides containing S121, S613 or S935 identified upon phospho-proteomic analysis of data shown in **f** and volcano plot of identified phosphosites with significant enrichment in red.
**j**, Pex11–GFP turnover in Δ*atg11* or Δ*atg11*Δ*pho81* cells harboring ev or expressing plasmid-encoded *ATG11* or *atg11$^{3SA}$* (S121A, S613A, S935A) after P-S (24 h). Data are mean ± s.d. (*n* = 6 biologically independent experiments). In **g**, **h** and **j**, statistical significance was assessed using one-way ANOVA followed by Tukey's multiple comparisons test. *P* values relative to WT (ev, *) or *PHO81* expressing Δ*pho81* cells (#) (**g** and **h**) or *ATG11*. *P* values *** or ###, *P* < 0.0001 except **P* = 0.0492 (**h**) and ***P* = 0.0037 (**j**), non-significant (n.s.). Source numerical data and unprocessed blots are available in Source data.

a Pho81$^{YKK}$ variant with mutated residues Y24A, K28A and K154A, predicted to be deficient for phospho-metabolite binding (Fig. 6a,b)[28]. CoIP–MS analysis demonstrated that changes in the SPX domain did not affect the interaction of Pho81 with the AKC or AKC components with each other upon P-S (Fig. 6c). However, the presence and functional integrity of the SPX domain was required for Pho81 to compensate for the absence of Atg13 during P-S-induced pexophagy (Fig. 6d). Importantly, metabolite sensing by the SPX domain of Pho81 regulated pexophagy in a manner that was independent of the PHO pathway (Extended Data Fig. 1d). In contrast to Pho81, fluorescence imaging showed GFP-tagged Pho81$^{ΔSPX}$ strongly accumulated at Atg1–mCherry puncta without forming cortical structures in cells independent of starvation (Fig. 6e), indicating a negative regulatory function of the SPX domain for AKC association of Pho81. Consistently, GFP-tagged Pho81$^{YKK}$ constitutively localized to Atg1 puncta (Fig. 6e). However, in contrast to Pho81$^{ΔSPX}$, Pho81$^{YKK}$ also formed cortical structures in the presence of phosphate (Fig. 6e), suggesting that, in addition to regulating AKC association, the SPX domain controls cortical localization of Pho81. Y2H assays demonstrated increased binding of AD-Pho81$^{ΔSPX}$ with BD-Atg11 compared with AD-Pho81 or AD-Pho81$^{YKK}$ (Fig. 6f), suggesting unrestrained binding of Pho81 to the AKC may inhibit pexophagy. Consistently, overexpressed Pho81 impaired pexophagy and generally blocked Atg11-dependent selective turnover of autophagic membranes in the absence of Atg13 (Fig. 6g,h). Taken together, these data establish a critical role for the phospho-metabolite sensing SPX domain in regulating the association of Pho81 with the AKC and promoting Pho81-dependent pexophagy.

The code for inositol polyphosphates InsP$_6$ and pyrophosphates InsP$_{7/8}$ signalling during P-S has been controversial[27,28,42] Recent work suggests declining concentrations of InsP$_8$ activate Pho81 and induce the PHO pathway[42]. To test whether phospho-metabolites regulate pexophagy, we examined pexophagy upon InsP$_6$ supplementation, the precursor for InsP$_8$ synthesis, during P-S. InsP$_6$ supplementation significantly reduced pexophagy in a partially Pho81-dependent manner (Fig. 6i). Additionally, overexpression of the inositol polyphosphate kinase Kcs1, which drives InsP$_8$ synthesis, resulted in a significant reduction in pexophagy (Fig. 6j)[28,42]. These data support a role for InsP$_8$ in pexophagy regulation. Taken together, our data support a model in which phospho-metabolite sensing via the conserved SPX domain regulates the binding of Pho81 to the AKC to tune pexophagy during P-S.

## Discussion

How cells convert metabolic information into stress-adapted autophagy responses is a key question. Our work describes a molecular mechanism in which a conditional subunit conveys metabolic information directly to the AKC. By comparing P-S and N-S in yeast, our work uncovers significant compositional and functional plasticity of the AKC in dependence of metabolic stress. In parts, this plasticity is based on the presence of a conditional subunit of the AKC, Pho81. Pho81 is critical to compensate for partially inactive Atg13 due to significant activity of TORC1 during P-S. Pho81 binds to Atg11 via an Atg11/FIP200 interaction motif at the AKC to regulate pexophagy upon P-S. Our work indicates that, rather than functioning as a canonical SAR in parallel with Atg36 and Pex3, Pho81 binds to the AKC to induce Atg11 phosphorylation probably by Atg1 kinase to promote pexophagy. Regulation of selective autophagy by Atg1-mediated Atg11 phosphorylation has been recently shown to affect receptor binding[16]. Our work now uncovers a molecular mechanism of how cells exploit Atg11 phosphorylation to tune selective autophagy according to metabolic inputs. Interestingly, Pho81-dependent phosphorylation sites in Atg11 are distinct from previously identified sites[16], implying a growing mechanistic scope of phospho-regulation of Atg11. Our data suggest phospho-metabolite sensing by the conserved SPX domain is required for Pho81 to compensate for the absence of Atg13 and to drive pexophagy. Our work supports a model in which Pho81 binds to the adaptor protein Atg11 as a modulator subunit to convey metabolic information directly to the AKC to regulate selective autophagy.

Based on our work, we propose that Atg11/FIP200 subunits bind two classes of proteins, conceptually categorized into receptors and modulators. Excitingly, our and work from others suggests the existence of a diverse class of modulator proteins of the AKC. For example, induction of xenophagy upon bacterial invasion in mammalian cells requires association of protein kinase TBK1 with FIP200, which is mediated by two adaptor proteins, NAP1/SINTBAD, both containing FIRs[35]. TNIP1 negatively regulates mammalian mitophagy in parts by binding to FIP200 via a potential FIR[45]. FIP200 also binds Atg16L1 via a FIR to promote starvation-induced autophagy in mammalian cells[46,47]. These few examples demonstrate remarkable versatility of modulator proteins in modifying activity and functions of the core AKC. We are probably only beginning to explore the scope of modulator subunits of Atg11/FIP200.

Our mechanistic work has the potential to translate into therapeutical approaches. In a mouse model, *Cryptococcus neoformans* expressing Pho81 variants with defects in their SPX domain fail to establish lung and brain infections due to their inability to sense inositol pyrophosphates[43]. Our work on the function of Pho81 as a metabolite sensing modulator of selective autophagy may provide mechanistic insights with potential for development of antifungal drugs[43,48].

## Online content

Any methods, additional references, Nature Portfolio reporting summaries, source data, extended data, supplementary information,

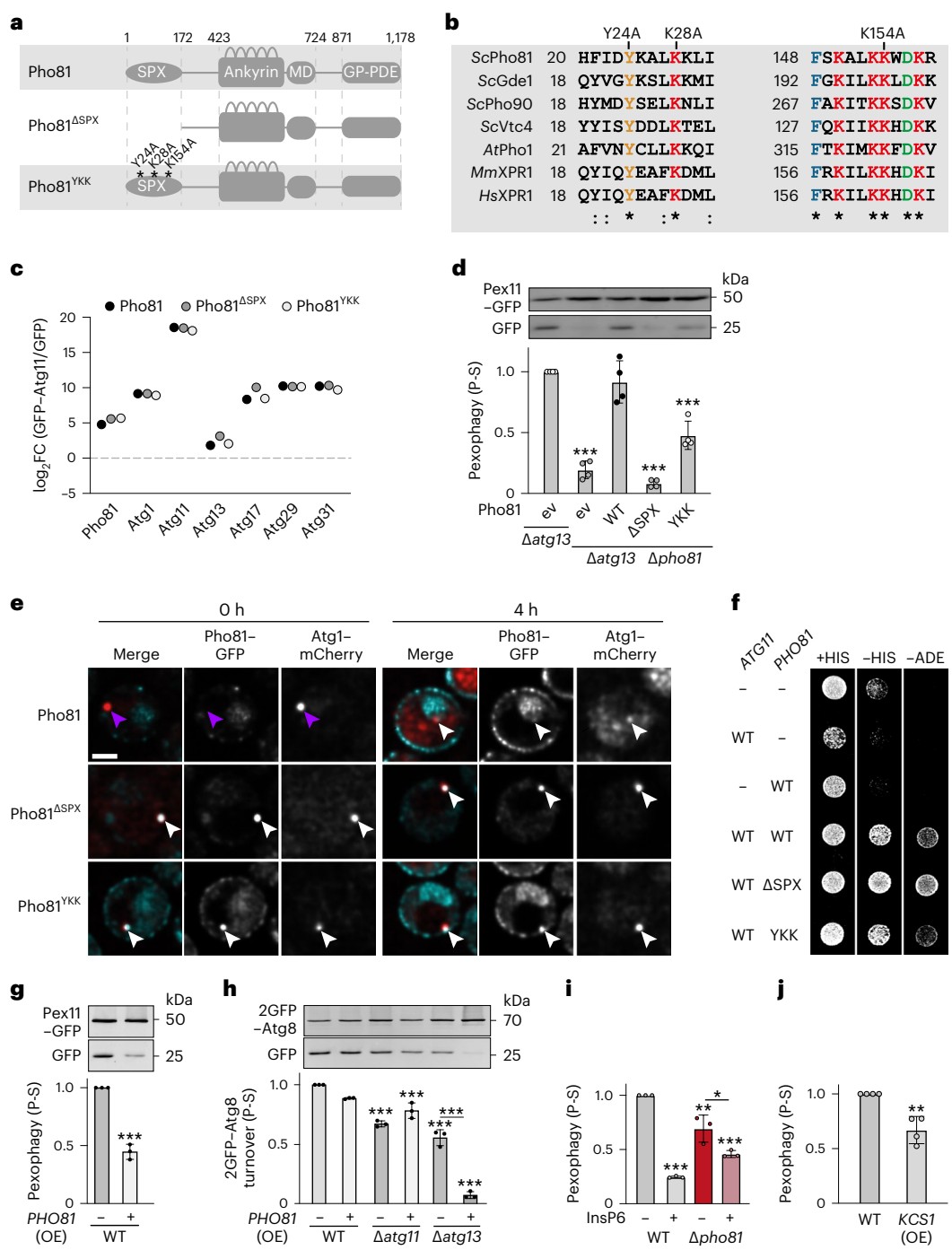

**Fig. 6 | SPX domain-dependent phospho-metabolite sensing by Pho81 regulates pexophagy. a**, Schematics of the Pho81 SPX domain variants; Pho81^{ΔSPX} (C-terminus 173–1,178); Pho81^{YKK}: Y24A, K28A and K154A indicate mutated residues involved in inositol phosphate sensing. **b**, Sequence alignment of SPX domain containing proteins from *Sc, Saccharomyces cerevisiae; At, Arabidopsis thaliana; Mm, Mus musculus; Hs, Homo sapiens*. **c**, Enrichment of AKC subunits after GFP–Atg11-based CoIP–MS from cells expressing plasmid-encoded *PHO81*–, *pho81^{YKK}*– or *pho81^{ΔSPX}*–*mCherry*. Control is GFP. *n* = 4 biologically independent experiments. Statistical analysis is presented in Supplementary Table 2. FC, fold change. **d**, Pex11–GFP turnover in strains harboring empty vector (ev) or expressing indicated plasmid-encoded *PHO81* variants in the respective *PEX11*–GFP-expressing deletion cells upon 24 h of P-S. Data are mean ± s.d. (*n* = 4 biologically independent experiments). **e**, Fluorescence imaging of cells expressing genomic *PHO81–GFP* variants and *ATG1–mCherry* during growth (0 h) and PS (4 h). Arrows indicate Pho81–GFP positive (white) or negative (purple) Atg1–mCherry foci. Data are representative of three biologically independent experiments. Scale bar, 2 μm. **f**, Y2H of cells expressing pGAD-*PHO81* or indicated

variants in combination with pGBDU-empty vector (−) or pGBDU-*ATG11* grown on +HIS, −HIS, −ADE selection plates. **g**, Pex11-GFP turnover in cells harboring ev (−) or expressing *PHO81* under an *ADH1* promoter (OE) and *PEX11–GFP* upon 24 h P-S. Data are mean ± s.d. (*n* = 3 biologically independent experiments). **h**, 2GFP-Atg8 turnover in indicated strains harboring ev (−) or expressing *PHO81* under an *ADH1* promoter (OE) after P-S (24 h). Data are mean ± s.d. (*n* = 3 biologically independent experiments). **i**, Pex11–GFP turnover in WT and Δ*pho81* cells after P-S (24 h) ± InsP6. Data are mean ± s.d. (*n* = 3 biologically independent experiments). **j**, Pex11–GFP turnover in cells harbouring pRS315 or pRS315–*prADH1-KCS1* (OE) after P-S (24 h). Data are mean ± s.d. (*n* = 4 biologically independent experiments). Statistical significance was assessed using one-way ANOVA followed by Tukey's multiple comparisons test (**d** and **i**), two-tailed *t*-test (**g** and **j**) and two-way ANOVA followed by Tukey's multiple comparisons test (**h**); *P* values are relative to WT (**g**–**j**), Δ*atg13* cells (**d**) or as indicated: ***P < 0.0001 except *P = 0.0096 and **P = 0.0018 (**i**) and **P = 0.0018 (**j**). Source numerical data and unprocessed blots are available in Source data.

acknowledgements; peer review information; details of author contributions and competing interests; and statements of data and code availability are available at https://doi.org/10.1038/s41556-024-01348-4.

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

## Methods

### Yeast strains and media

The strains used in this study were derivates of W303 or, for Y2H, derivates of PJ69-4a. All strains are listed in Supplementary Table 1. Gene deletions were generated by replacing the complete open reading frame with indicated marker cassettes using polymerase chain reaction-based targeted homologous recombination as described previously[51]. Plasmids were constructed by in vivo 'gap repair' based on homologous recombination into HindIII- and EcoRI-linearized pRS315 and pRS316 or BamHI-EcoRI-linearized pGAD and pGBDU, respectively[52]. Gene modification for C-terminal tagging used pFA6a–*link–yEGFP–CaURA3*, pFA6a–*link–3yEGFP–CaURA3*, pFA6a–*link–mCherry–kanMX6* or pFA6a–*link–mCherry–HIS3* as previously described[53,54]. Strains were grown at 30 °C in synthetic complete medium SDC (0.7% (w/v) yeast nitrogen base (BD Difco, 291920), 2% (w/v) α-D-glucose (Sigma, G8270) and a supplement of all required amino acids, adenine and uracil) or synthetic medium lacking respective amino acids for plasmid selection. For starvation experiments, cells were collected from early log phase ($OD_{600}$ 0.5–1), washed four times and resuspended in respective starvation media. For P-S, synthetic medium lacking phosphate (SD-Pi) medium contained 0.15% (w/v) yeast nitrogen base without amino acids, ammonium sulfate and phosphate, supplemented with KCl (Formedium, CYN6802), 2% (w/v) α-D-glucose (Sigma, G8270), 0.5% (w/v) ammonium sulfate (Sigma, A4418) and a supplement of all required amino acids, adenine (Sigma, A9126) and uracil (Sigma, U0750). For N-S, SD-N medium contained 0.17% (w/v) yeast nitrogen base without amino acids and ammonium sulfate (BD Difco, 233520) and 2% (w/v) α-D-glucose (Sigma, G8270). When indicated, rapamycin was added at a final concentration of 400 ng ml$^{-1}$ in dimethyl sulfoxide (Calbiochem, 553210). For $InsP_6$ supplementation, a final concentration of 250 μg ml$^{-1}$ of phytic acid sodium salt hydrate (Sigma, 68388) was added to SD-Pi medium.

### Y2H assay

For Y2H assays, 1 $OD_{600}$ unit of indicated strains was serially diluted (1:10) and four concentrations were spotted on corresponding selection plates (SD or SD-Pi medium containing 2% (w/v) agar (MP, 100262) and defined amino acid compositions). Plates were imaged after 4 days of growth at 30 °C with the Bio-Rad Chemidoc IP Imaging System.

### Fluorescence microscopy and image analysis

Cells were transferred to 96-well glass-bottom microplates (Greiner Bio-One) containing indicated media. For fixed timepoint analyses, cells were imaged at room temperature with an inverted microscope (Nikon Ti-E) using a Plan Apochromat IR 60× 1.27 numerical aperture objective (Nikon) and Spectra X LED light source (Lumencor). Three-dimensional light microscopy data were collected using the triggered Z-Piezo system (Nikon) and orca flash 4.0 camera (Hamamatsu). High-speed confocal imaging was performed with a Dragonfly 500 series spinning disk microscope (Andor, Oxford Instruments) equipped with a Zyla 4.2 Plus sCMos camera (Andor) using a Lamba CFI-Plan Apochromat 60× 1.4 numerical aperture oil immersion objective (Nikon). Three-dimensional data were processed using Fusion software (Andor), and Fiji ImageJ Version 2.1.0. Deconvolution was performed using Huygens Professional 16.10 where indicated. Data are shown as single sections if not stated otherwise.

### Whole cell extraction, western blot analysis and quantification

A total of 0.5 $OD_{600}$ units of cells were collected and lysed by alkaline whole cell extraction using 0.255 M NaOH (Merck, 011-002-00-6). Proteins were precipitated with 50% (w/v) trichloroacetic acid (Roth, 8789.2) and washed once with ice-cold acetone (Merck, 179124). Protein pellets were resuspended in 1× sodium dodecyl sulfate (SDS, Roth, CN30.2) sample buffer (50 mM Tris/HCl pH 6.8 (Roth, 4855.3), 10% (v/v) glycerol (VWR, 24388364), 1% (w/v) SDS (Roth, CN30.2), 0.01% (w/v)

bromphenolblue (Roth, A512.1) and 1% (v/v) β-mercaptoethanol (Roth, 4227.3)). Protein extracts were analysed by SDS polyacrylamide gel electrophoresis and immunoblotting using α-GFP (JL-8, monoclonal, Takara, 1:1,000 in 5% milk powder in TBS-T), α-mCherry (polyclonal, Genetex, 1:2,000 in 5% milk powder in TBS-T) and α-Atg13 (polyclonal, gift from Prof. Daniel Klionsky, 1:10,000 in 5% milk powder in TBS-T) antibodies. Primary antibodies were visualized using corresponding secondary α-mouse or α-rabbit Dylight800/680 antibodies 1:10,000 (Rockland Immunochemicals), and signal intensities were quantified using the Li-COR Odyssey Infrared Imaging system (Biosciences) and analysed using ImageStudioLite (Version 5.2.5). For GFP-based autophagy flux assays, the intensities for free GFP were divided by the total intensities of free GFP + full-length protein–GFP. Data are shown as absolute ratios or expressed as relative values after normalization to the reference conditiona set as 1.

### Whole cell extraction for MS analysis

One $OD_{600}$ unit of cells was collected in 2.0 ml safe lock tubes (Eppendorf, 0030 121.880), washed once with $H_2O$ and snap frozen in liquid nitrogen. Cells were disrupted using Tissue Lyser (Qiagen) for 1 min at 25 Hz. One millilitre −20 °C cold Extraction Buffer MTBE:MeOH 75:25 (v/v) (Sigma, 306975; Biosolve, 136841) was added, and samples were vortexed thoroughly, ultrasonicated for 10 min in bath-type sonicator cooled with ice and incubated on a thermomixer at 4 °C for 30 min, and then centrifuged for 10 min at 4 °C and 16,000g. Supernatants were removed and pellets (containing proteins) dried under the hood and stored at −80 °C. The pellet was resolved in 50 μl urea (8 M, Sigma, U5378) and 0.4 μl tris(2-carboxyethyl)phosphine (0.25 M, Sigma, 75259) and 0.69 μl chloracetic acid (0.8 M) were added before 1 h incubation at room temperature. After addition of 1 μl Lys-C (0.5 μg μl$^{-1}$) (Endoproteinase, Life Technologies, 90051) samples were incubated for >2 h at room temperature. Then 150 μl ammonium bicarbonate (50 mM, Sigma, 09830) was added followed by incubation with 1 μl trypsin (1 μg μl$^{-1}$, Trypsin-gold Promega, V5280) over night at 37 °C. Reactions were stopped by the addition of 0.1% (v/v) trifluoroacetic acid (Sigma, 302031).

### Co-immunoprecipitations

A tota of 125 $OD_{600}$ units (for subsequent western blot analysis) or 250 $OD_{600}$ units (for MS-coupled analysis) of cells were pelleted, washed and resuspended in lysis buffer (50 mM Tris/HCl pH 7.5 (Roth, 4855.3), 150 mM NaCl (Fluka, 31434), 1 mM $MgCl_2$ (Roth, KK36.2), 10% (v/v) glycerol (VWR, 24388364), Complete protease inhibitor (Roche, 04693132001), and 1 mM phenylmethylsulfonyl fluoride (Sigma, P7626) or 2 mM AEBSF-hydrochloride (BioChemica, APA1421.0001)). Cell pellets were created in liquid $N_2$ and stored at −80 °C until further processing. Frozen cell pellets were lysed using a Freezer/Mill (6970EFM, SPEX Metuchen, NJ; settings: T1, 5 min; T2, 2 min; T3, 2 min; cycles, 2; rate, 7). Thawed cell lysates were cleared by two consecutive centrifugation steps at 2,110g (Heraeus Multifuge X3R, TX-1000 rotor, Thermo Scientific) and 13,500g (Eppendorf Centrifuge 5424) at 4 °C for 10 min each. After clearance, 0.25% NP-40 (Milipore, 492016) was added to supernatants. Supernatants were incubated with μMACS anti-GFP Microbeads (Miltenyi Biotec, 130-094-252) for 1 h at 4 °C. Beads were isolated using Miltenyi μ-columns and a MultiMACS M96 Seperator (Miltenyi Biotec). Columns were equilibrated with lysis buffer, loaded and washed twice with washing buffer 1 (50 mM Tris/HCl pH 7.5 (Roth, 4855.3); 150 mM NaCl (Fluka, 31434); 10% glycerol (VWR, 24388364); 0.25% NP-40 alternative (Sigma, 492016) and twice with washing buffer 2 (50 mM Tris/HCl pH 7.5 (Roth, 4855.3), 150 mM NaCl (Fluka, 31434) and 10% glycerol (VWR, 24388364)).

For analysis by western blotting, samples were eluted by addition of 100 μl SDS-sample buffer. For analyses by MS, on-bead digestion was performed by addition of 100 μl trypsinization buffer (50 mM ammonium bicarbonate (Sigma, 09830) and 300 ng trypsin per column (Trypsin-gold, Promega, V5280)) for 1 h at 23 °C. Samples were

eluted by addition of 150 µl 50 mM ammonium bicarbonate (Sigma, 09830). After overnight incubation at 23 °C, reactions were stopped by the addition of 0.1% (v/v) trifluoroacetic acid (Sigma, 302031). The aqueous solution was evaporated in a SpeedVac (Eppendorf) and analysed by nano-electrospray ionization–tandem MS (MS/MS) analysis or high-resolution MS.

## CoIP proteomics analysis

Peptides were recovered and separated on a 25-cm, 75-µm-internal-diameter PicoFrit analytical column (New Objective) packed with 1.9 µm ReproSil-Pur 120 C18-AQ medium (Dr. Maisch), using an EASY-nLC 1200 (Thermo Fisher Scientific). The column was maintained at 50 °C. Buffers A and B were 0.1% formic acid (Sigma, 33015) in water and 0.1% formic acid in 80% acetonitrile (ACN; Sigma, 34998). Peptides were separated on a segmented gradient from 6% to 31% buffer B for 40 or 80 min and from 31% to 50% buffer B for 5 or 10 min at 200 nl min$^{-1}$. Eluting peptides were analysed on a QExactive HF mass spectrometer (Thermo Fisher Scientific). Peptide precursor $m/z$ measurements were carried out at 60,000 or 12,000 resolution in the 300 to 1,800 $m/z$ range. The top ten most intense precursors with charge state from 2 to 7 only were selected for higher-energy collisional dissociation fragmentation using 25% normalized collision energy. The $m/z$ values of the peptide fragments were measured at a resolution of 30,000 using a minimum automatic gain control (AGC) target of 8e3 and 80 ms maximum injection time. Upon fragmentation, precursors were put on a dynamic exclusion list for 20 or 45 s.

## CoIP phosphoproteomics

Peptides were recovered and separated on a 40-cm, 75-µm-internal-diameter packed emitter column (Coann emitter from MS Wil, Poroshell EC C18 2.7 µm medium from Agilent) using an EASY-nLC 1000 (Thermo Fisher Scientific). The column was maintained at 50 °C. Buffers A and B were 0.1% formic acid (Sigma, 33015) in water and 0.1% formic acid in 80% ACN (Sigma, 34998). Peptides were separated on a segmented gradient from 6% to 31% buffer B for 57 min 300 nl min$^{-1}$ followed by a high organic wash phase and an additional short gradient for column cleaning. Eluting peptides were analysed on an Orbitrap Fusion mass spectrometer (Thermo Fisher Scientific). Peptide precursor $m/z$ measurements were carried out at 60k resolution in the 350 to 1,500 $m/z$ range. Precursors with charge state from 2 to 7 were selected for higher-energy collisional dissociation fragmentation (normalized collision energy 27) in a Topspeed method with a cycle time of 1 s. Fragment ion spectra were measured in the Orbitrap analyser at a resolution of 50k with an AGC target of 2e5 and a maximum injection time of 86 ms. Isolated precursors were excluded from further fragmentation for 45 s.

## Whole cell proteomics analysis

For the analysis of the total proteome, 4 µg of desalted peptides were labelled with tandem mass tags (TMT10plex, Thermo Fisher Scientific, 90110) using a 1:20 ratio of peptides to TMT reagent[55]. TMT labelling was carried out according to the manufacturer's instruction with the following changes: dried peptides were reconstituted in 9 µl 0.1 M tetra-ethylammonium bromide to which 7 µl TMT reagent in ACN (Sigma, 34998) was added to a final ACN concentration of 43.75%, the reaction was quenched with 2 µl 5% hydroxylamine (Sigma, 159417). Labelled peptides were pooled, dried, resuspended in 0.1% formic acid (Sigma, 33015), split into two samples, and desalted using home-made STAGE tips. One of the two samples was fractionated on a 1 mm × 150 mm ACQUITY column, packed with 130 Å, 1.7 µm C18 particles (Waters, SKU: 186006935), using an Ultimate 3000 UHPLC (Thermo Fisher Scientific). Peptides were separated at a flow of 30 µl min$^{-1}$ with a 96 min segmented gradient from 1% to 50% buffer B for 85 min and from 50% to 95% buffer B for 11 min; buffer A was 5% ACN, 10 mM ammonium bicarbonate (ABC, Sigma, 09830), and buffer B was 80% ACN, 10 mM ABC. Fractions were collected every 3 min, and fractions were pooled

in two passes (1 + 17, 2 + 18 and so on) and dried in a vacuum centrifuge (Eppendorf). Dried fractions were resuspended in 0.1% formic acid and analysed on a Orbitrap Lumos Tribrid mass spectrometer (Thermo Fisher Scientific) equipped with a FAIMS device (Thermo Fisher Scientific) that was operated in two compensation voltages, −50 and −70. Synchronous precursor selection-based MS3 was used for TMT reporter ion signal measurements. Raw files were split on the basis of the FAIMS compensation voltage using FreeStyle (Thermo Fisher Scientific).

## Protein identification and quantification

The raw data were analysed with MaxQuant version 1.5.3.17, 1.6.10.43 or 2.3.0 using the integrated Andromeda search engine[56]. Peptide fragmentation spectra were searched against the yeast proteome (downloaded May 2017, February 2018 or October 2018 from UniProt). Methionine oxidation, protein N-terminal acetylation and phosphorylation on S, T and Y were set as variable modifications; cysteine carbamidomethylation was set as fixed modification. The digestion parameters were set to 'specific' and 'Trypsin/P'; the minimum number of peptides and razor peptides for protein identification was 1; the minimum number of unique peptides was 0. Protein identification was performed at a peptide spectrum matches and protein false discovery rate (FDR) of 0.01. For the analysis of the total proteome samples, the isotope purity correction factors, provided by the manufacturer, were included in the analysis. The 'second peptide' option was on. Successful identifications were transferred between the different raw files using the 'Match between runs' option. Label-free quantification (LFQ) was performed using an LFQ minimum ratio count of 2. LFQ intensities were filtered for at least four valid values in at least one group and imputed from a normal distribution with a width of 0.3 and down shift of 1.8. Differential expression analysis was performed using limma[49], version 3.34.9 in R, version 3.4.3 (R Core Team 2017).

## LC–MS/MS measurements and data analysis for phosphoproteomics

MS sample preparation and liquid chromatography (LC)–MS/MS analyses were performed as described[7]. Briefly, proteins were extracted by disrupting yeast cells in 8 M urea (Sigma, U5378) using a bead beater, and digested by Lys-C (Thermo Scientific, 90051) for 4 h at room temperature. Before overnight digestion, the concentration of urea was diluted to 1 M. The next day, peptides were purified by solid phase extraction using HR-X columns. Dry peptides were suspended in 200 µl 80% ACN (Sigma, 34998) with 1% trifluoroacetic acid (Sigma, 302031) for phosphopeptide enrichment using Fe(III)-NTA cartridges on an AssayMAP Bravo (Agilent). The flow-through was stored at −80 °C for non-phosphopeptide analysis. LC–MS/MS measurements were performed on an Exploris 480 mass spectrometer coupled to an EasyLC 1200 nanoflow-HPLC or Vanquish Neo UHPLC System (all Thermo Scientific). Peptides were separated on a fused silica HPLC-column tip (inner diameter 75 µm, New Objective, self-packed with ReproSil-Pur 120 C18-AQ, 1.9 µm (Dr. Maisch) to a length of 20 cm) using a gradient of A (0.1% formic acid in water) and B (0.1% formic acid in 80% ACN in water). For non-phosphopeptide analyses, mass spectrometer was operated in the data-dependent mode; after each MS scan (mass range $m/z$ = 370–1,750; resolution: 120,000) a maximum of 20 MS/MS scans were performed using an isolation window of 1.3, a normalized collision energy of 28%, a target AGC of 50% and a resolution of 15,000. Spray voltage was set to 2.3 kV and the ion-transfer tube temperature to 250 °C; no sheath and auxiliary gas were used. MS raw files were analysed using MaxQuant (version 2.0.1.0)[56] using a UniProt full-length *Saccharomyces cerevisiae* database (March, 2016) and common contaminants, such as keratins and enzymes used for in-gel digestion, as reference. Carbamidomethylcysteine was set as fixed modification, and protein amino-terminal acetylation, serine, threonine and tyrosine phosphorylation, and oxidation of methionine were set as variable modifications. The MS/MS tolerance was set to 20 ppm, and three missed cleavages were allowed using trypsin/P as enzyme

specificity. Peptide, site and protein FDR based on a forward-reverse database were set to 0.01, minimum peptide length was set to 7, the minimum score for modified peptides was 40, and minimum number of peptides for identification of proteins was set to 1, which must be unique. Phosphopeptides were analysed on the same instruments in data independent mode in which a full-scan MS (scan range: 350–1,200 $m/z$ with a resolution of 120,000; target AGC of 300%; maximum injection time of 60 ms) was followed by MS/MS with 30.4 $m/z$ isolation window with 1 Da overlap over a range of 350–1,200 $m/z$ (target AGC of 300%; resolution 30,000; stepped normalized collision energy of 25.5, 27 and 30). MS raw files were analysed using Spectronaut software (version 17) using the default setting and a spectral library of phosphopeptides generated by data-dependent acquisition.

LC–MS/MS data analysis was performed by Perseus software[57]. Briefly, phosphosite intensities were normalized to respective protein intensities followed by the median intensity of each biological replicate to discriminate regulated sites from regulated proteins ($n = 3$ per treatment and strain). Sites were filtered on a minimum of six valid values and a minimal localization probability of ≥0.75 (class I sites)[58]. To identify significantly different sites between strains and treatments, pairwise comparisons using Student's $t$-tests were performed (permutation-based FDR <0.05). Replicate experiments were combined by calculating the median and for hierarchical clustering using $k$-means pre-processing intensities were normalized using $z$-scores. Sites that changed significantly in minimal one pairwise comparison were filtered on known TORC1 and Atg1 target sites using datasets published in ref. 7.

## Statistics and reproducibility
Data are independent biological replicates. No statistical method was used to pre-determine sample size. No data were excluded from the analyses. The experiments were not randomized. Except for MS-based analysis, the investigators were not blinded to allocation during experiments and outcome assessment. For western blot, imaging, Y2H and proteomics assays, quantifications and statistics were derived from $n = 3$ independent experiments unless specified in the legends. Error bars represent standard deviation (s.d.) as indicated in the figure legends. Box-and-whiskers plots: the boxes extend from the 25th to the 75th percentile spanning the interquartile range (IQR), the whiskers show the minimal and the maximal value (minimum to maximum), and the line in the middle of the boxes represents the median. Data were processed and analysed in GraphPad Prism 8.4.3. Statistical analysis was performed using an ordinary one-way analysis of variance (ANOVA) with Tukey's multiple comparisons test of one independent variable or a two-way ANOVA with Šidák's multiple comparisons test of two independent variables, as indicated ($^*P < 0.05$, $^{**}P < 0.01$, $^{***}P < 0.001$). Figures and graphs were assembled in Adobe Illustrator Version 24.3. Protein multiple sequence alignments were performed using Clustal Omega (1.2.4)[59].

## Reporting summary
Further information on research design is available in the Nature Portfolio Reporting Summary linked to this article.

## Data availability
Mass spectrometry data have been deposited in ProteomeXchange with the primary accession code PXD045932 (Fig. 3a–c) and PXD048177 (Figs. 1c, 2a,b, 5f,i and 6c). Mass spectrometry datasets have also been deposited to figshare and can be accessed at the following links: https://doi.org/10.6084/m9.figshare.21718160; Fig. 1c (https://doi.org/10.6084/m9.figshare.21751184), Fig. 2a (https://doi.org/10.6084/m9.figshare.21751187), Fig. 2b (https://doi.org/10.6084/m9.figshare.24450133), Fig. 5f,i (https://doi.org/10.6084/m9.figshare.21751190) and Fig. 6c (https://doi.org/10.6084/m9.figshare.21751181). Source data are provided with this paper. All other data supporting the findings of this study are available from the corresponding author upon reasonable request.

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

## Acknowledgements
We thank I. Huppertz for generous support and L. Pernas, S. Hofer and all members of the Graef group for discussion; D. Klionsky for the α-Atg13 primary antibody; and the members of the Proteomics Core and FACS & Imaging Facility at the Max Planck Institute for Biology for excellent support. This work was supported by the Swiss National Science Foundation (SNSF) project number 310030 to J.D., and the Max Planck Society and the Deutsche Forschungsgemeinschaft (DFG—German Research Foundation)—SFB 1218—project number 269925409 to M.G.

## Author contributions
A.S.G., R.G. and M.G. conceptualized the study; A.S.G., R.G., M.Sc., E.R., A.R., B.C.B., M.St., J.D. and M.G. designed and performed experiments and analysed and interpreted the data; A.S.G. and M.G. wrote the manuscript. All authors approved the final version of the manuscript.

## Funding

## Competing interests
The authors declare no competing interests.

## Additional information
**Extended data** is available for this paper at https://doi.org/10.1038/s41556-024-01348-4.

**Correspondence and requests for materials** should be addressed to M. Graef.

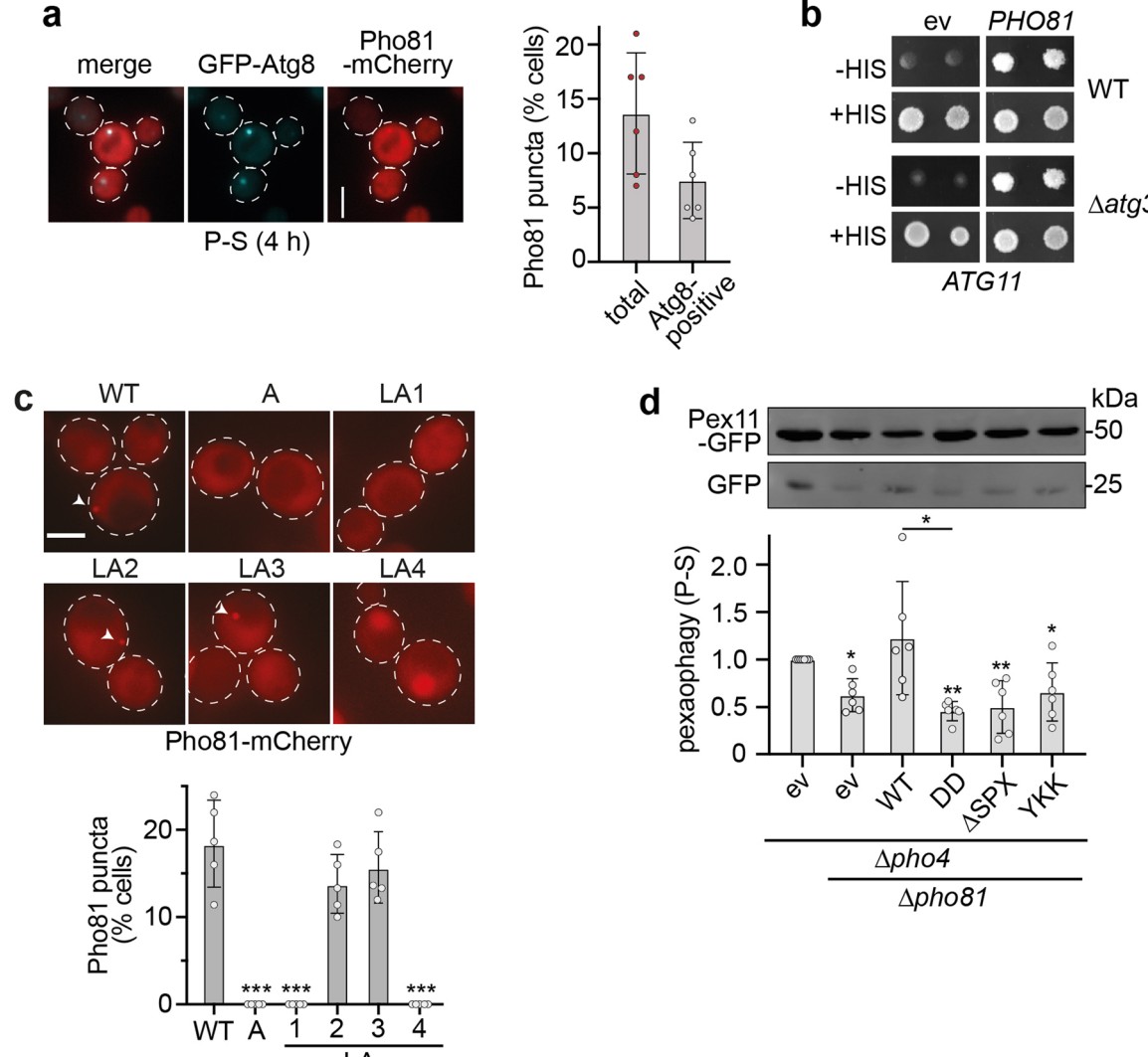

**Extended Data Fig. 1 | Additional analysis of Pho81 in autophagy during P-S.**
**a**, Fluorescence imaging of strains co-expressing plasmid encoded *PHO81*-mCherry under *ADH1* promoter and genomic 2GFP-*ATG8* upon phosphate starvation (4 h). Scale bar = 5 µm. Quantification of Pho81-mCherry puncta and co-localization with 2GFP-Atg8. Data are means ± SD (n = 100 cells examined over 5 independent experiments). **b**, Y2H of cells harboring pGAD (ev) or pGAD-*PHO81* and pGBDU-*ATG11* in WT (pJ69-4a) or *Δatg36* cells grown on SD + HIS or SD-HIS media. **c**, Fluorescence imaging of strains expressing plasmid-encoded *PHO81*-mCherry variants under *ADH1* promoter during growth. Scale bar = 5 µm. Quantification

of Pho81-mCherry puncta per cell. Data are means ± SD (n = 125 cells examined over 5 independent experiments). **d**, Pex11-GFP turnover in cells expressing indicated plasmid-encoded *PHO81* variants in *Δpho4* or *Δpho4Δpho81* cells after phosphate starvation (24 h). Data are normalized to the empty vector control in *Δpho4* cells and are means ± SD (n = 6 biologically independent experiments). Statistical significance was assessed using one-way ANOVA followed by Tukey's multiple comparison test. P-values are relative to WT (c), ev control in *Δpho4*, or as indicated: *, p < 0.05; **, p < 0.01; ***, p < 0.001. Source numerical data and unprocessed blots are available in source data.

# Reporting Summary

## Statistics

For all statistical analyses, confirm that the following items are present in the figure legend, table legend, main text, or Methods section.

| n/a | Confirmed | |
|---|---|---|
| ☐ | ☒ | The exact sample size (*n*) for each experimental group/condition, given as a discrete number and unit of measurement |
| ☐ | ☒ | A statement on whether measurements were taken from distinct samples or whether the same sample was measured repeatedly |
| ☐ | ☒ | The statistical test(s) used AND whether they are one- or two-sided<br>*Only common tests should be described solely by name; describe more complex techniques in the Methods section.* |
| ☒ | ☐ | A description of all covariates tested |
| ☐ | ☒ | A description of any assumptions or corrections, such as tests of normality and adjustment for multiple comparisons |
| ☐ | ☒ | A full description of the statistical parameters including central tendency (e.g. means) or other basic estimates (e.g. regression coefficient) AND variation (e.g. standard deviation) or associated estimates of uncertainty (e.g. confidence intervals) |
| ☐ | ☒ | For null hypothesis testing, the test statistic (e.g. *F*, *t*, *r*) with confidence intervals, effect sizes, degrees of freedom and *P* value noted<br>*Give P values as exact values whenever suitable.* |
| ☒ | ☐ | For Bayesian analysis, information on the choice of priors and Markov chain Monte Carlo settings |
| ☒ | ☐ | For hierarchical and complex designs, identification of the appropriate level for tests and full reporting of outcomes |
| ☒ | ☐ | Estimates of effect sizes (e.g. Cohen's *d*, Pearson's *r*), indicating how they were calculated |

*Our web collection on statistics for biologists contains articles on many of the points above.*

## Software and code

Policy information about availability of computer code

| Data collection | *Provide a description of all commercial, open source and custom code used to collect the data in this study, specifying the version used OR* |
|---|---|
| Data analysis | Proteomics raw data was analzyed using MaxQuant (1.5.3.17, 1.6.10.43, 2..0.1.0, or 2.3.0) using the integrated Andromeda search engine. Imaging data was processed using Fusion software (Andor) and Fiji ImageJ Version 2.1.0. Deconvolution was performed using Huygnes Professional 16.10. Western blot data was analyzed using ImageStudioLite Version 5.2.5. Sequence alignments were performed using Clustal Omega (1.2.4) |

For manuscripts utilizing custom algorithms or software that are central to the research but not yet described in published literature, software must be made available to editors and reviewers. We strongly encourage code deposition in a community repository (e.g. GitHub). See the Nature Portfolio guidelines for submitting code & software for further information.

## Data

Policy information about availability of data

All manuscripts must include a data availability statement. This statement should provide the following information, where applicable:
- Accession codes, unique identifiers, or web links for publicly available datasets
- A description of any restrictions on data availability
- For clinical datasets or third party data, please ensure that the statement adheres to our policy

The proteomics datasets shown in Fig. 1c (10.6084/m9.figshare.21751184), Fig. 2a (10.6084/m9.figshare.21751187), Fig. 2b (10.6084/m9.figshare.24450133), Fig.3 g-i (PXD045932), Fig. 4f and 4i (10.6084/m9.figshare. 21751190), Fig. 5c (10.6084/m9.figshare. 21751181) are available in the figshare repository.

## Human research participants

Policy information about studies involving human research participants and Sex and Gender in Research.

| | |
|---|---|
| Reporting on sex and gender | *not applicable* |
| Population characteristics | *not applicable* |
| Recruitment | *not applicable* |
| Ethics oversight | *not applicable* |

Note that full information on the approval of the study protocol must also be provided in the manuscript.

# Field-specific reporting

Please select the one below that is the best fit for your research. If you are not sure, read the appropriate sections before making your selection.

☒ Life sciences  ☐ Behavioural & social sciences  ☐ Ecological, evolutionary & environmental sciences

For a reference copy of the document with all sections, see nature.com/documents/nr-reporting-summary-flat.pdf

# Life sciences study design

All studies must disclose on these points even when the disclosure is negative.

| | |
|---|---|
| Sample size | A minimum of four (4) independent biological replicates were generated, except for the whole cell proteomics in Fig. 2a and the Co-IP MS in Fig 4f (n=3-4).  A  sufficient sample size was determined based on variance between experimental groups. |
| Data exclusions | Data were excluded from analysis only upon clear technical failure. |
| Replication | All experiments were replicated at a minimum of 4 independent biological replicates as indicated in figure legends. |
| Randomization | Proteomics samples were analyzed in randomized order (Fig. 1c, 2a, 4f, and 5c)<br>Analysis of imaging data was performed by randomly choosing cells from each replicate (Fig. 1e,f, 2f,g, 5e, Extended data 1a,c,d, e)<br>All other samples were analyzed based on genotypes, treatments, and/or time points with internal controls with randomization.<br>During fluorescence imaging, groups were imaged in random order to exclude positional effects in the imaging pipeline |
| Blinding | Mass spectrometry measurements and data analysis was performed and analyzed in a blinded manner. Fluorescence imaging was performed and analyzed in non-blinded manner, but independently verified by different experimentators. |

# Reporting for specific materials, systems and methods

We require information from authors about some types of materials, experimental systems and methods used in many studies. Here, indicate whether each material, system or method listed is relevant to your study. If you are not sure if a list item applies to your research, read the appropriate section before selecting a response.

## Materials & experimental systems

| n/a | Involved in the study |
|---|---|
| ☐ | ☒ Antibodies |
| ☒ | ☐ Eukaryotic cell lines |
| ☒ | ☐ Palaeontology and archaeology |
| ☒ | ☐ Animals and other organisms |
| ☒ | ☐ Clinical data |
| ☒ | ☐ Dual use research of concern |

## Methods

| n/a | Involved in the study |
|---|---|
| ☒ | ☐ ChIP-seq |
| ☒ | ☐ Flow cytometry |
| ☒ | ☐ MRI-based neuroimaging |

## Antibodies

| Antibodies used | Primary antibodies: α-GFP (monoclonal, Takara, 632380), α-mCherry (polyclonal, Genetex, GTX128508), α-Atg13 (polyclonal, a gift from Prof. Daniel Klionsky) antibodies.<br>Secondary antibodies: α-mouse or α-rabbit Dylight800/680 antibodies (Rockland Immunochemicals, 610-745-124 and 611-142-122 respectively). a–GFP (JL–8, monoclonal, Takara, 632380) |
|---|---|
| Validation | α-GFP (monoclonal, Takara, 632380) https://www.takarabio.com/documents/Certificate%20of%20Analysis/632380/632380-632381-070313.pdf<br>α-mCherry (polyclonal, Genetex, GTX128508) https://www.genetex.com/PDF/Download?catno=GTX128508<br>α-Atg13 doi: 10.1074/jbc.M002813200 and doi: 10.4161/auto.27707 |

