## [Peer Review File · Nature Cell Biology]

Peer Review Information

Journal: Nature Cell Biology

Manuscript Title: A metabolite sensor subunit of the Atg1/ULK complex regulates selective autophagy

Corresponding author name(s): Professor Martin Graef

Editorial Notes:

Reviewer Comments & Decisions:

Decision Letter, initial version:
--

Dear Professor Graef,

Thank you very much for submitting your manuscript, "A metabolite sensor subunit of the Atg1/ULK complex regulates selective autophagy", to Nature Cell Biology and thank you for your patience with the peer review process. The manuscript has now been seen by 3 referees, who are experts in autophagy initiation (Referee #1); autophagy, yeast (Referee #2); and pexophagy (Referee #3). As you will see from their comments (attached below), they found this work of potential interest but have raised substantial concerns, which in our view would need to be addressed with considerable revisions before we can consider publication in Nature Cell Biology.

As per our standard editorial process, we have now discussed the referee reports in detail within the editorial team, including the Chief Editor, to identify key referee points that should be addressed with priority. To guide the scope of the revisions, I have listed these points below. Our typical revision period is six months, and we are committed to providing a fair and constructive peer-review process, so please feel free to contact me if you would like to discuss any of the referee comments further or if you anticipate any issues or delays addressing the reviews.

You will see that the referees' concerns point to a premature dataset and concerns regarding the mechanism of action of Pho81 and its functional importance would need to be addressed with experiments and data, and reconsideration of the study for this journal and re-engagement of referees would depend on strength of these revisions:

(i) Refining our understanding of the mechanism of action of Pho81 has been requested by all reviewers, and we agree that added mechanistic insight would strengthen the advance despite the already significant body of work provided in the submission. We recommend that you address the

reviewers' questions to provide further support for the model that Pho81 may compensate for Atg13 function and dissect how Pho81 activates the Atg1 complex:

Rev#1 points #2, #3, #4, #6

Rev#2 major point #1

Rev#3 point F #3

We note that the reviewers all provide helpful suggestions to delineate the mechanism. While we feel the points above should be addressed rigorously, in our view, determining whether Pho81 undergoes LLPS with detailed analyses of this question is a lower priority than the other points, which are all important questions.

(ii) further assay the role of Pho81 across types of autophagy, with additional assays to add support to the conclusions:

Rev#1 points #1, #5

Rev#2 major point #2

Rev#3 points F #1-2

(iii) Finally, please pay close attention to our guidelines on statistical and methodological reporting (listed below) as failure to do so may delay the reconsideration of the revised manuscript. In particular please provide:

We would be happy to consider a revised manuscript that would satisfactorily address these points, unless a similar paper is published elsewhere, or is accepted for publication in Nature Cell Biology in the meantime.

- ensure that it conforms to our format instructions and publication policies (see below and <https://www.nature.com/nature/for-authors>).

- provide a point-by-point rebuttal to the full referee reports verbatim, as provided at the end of this letter.

- provide the completed Reporting Summary (found here <https://www.nature.com/documents/nr-reporting-summary.pdf>). This is essential for reconsideration of the manuscript will be available to editors and referees in the event of peer review. For more information

see <http://www.nature.com/authors/policies/availability.html> or contact me.

When submitting the revised version of your manuscript, please pay close attention to our [href="https://www.nature.com/nature-portfolio/editorial-policies/image-integrity">Digital Image Integrity Guidelines](https://www.nature.com/nature-portfolio/editorial-policies/image-integrity). and to the following points below:

Nature Cell Biology is committed to improving transparency in authorship. As part of our efforts in this direction, we are now requesting that all authors identified as 'corresponding author' on published papers create and link their Open Researcher and Contributor Identifier (ORCID) with their account on the Manuscript Tracking System (MTS), prior to acceptance. ORCID helps the scientific community achieve unambiguous attribution of all scholarly contributions. You can create and link your ORCID from the home page of the MTS by clicking on 'Modify my Springer Nature account'. For more information please visit www.springernature.com/orcid.

This journal strongly supports public availability of data. Please place the data used in your paper into a public data repository, or alternatively, present the data as Supplementary Information. If data can only be shared on request, please explain why in your Data Availability Statement, and also in the correspondence with your editor. Please note that for some data types, deposition in a public repository is mandatory - more information on our data deposition policies and available repositories appears below.

[Redacted]

We hope that you will find our referees' comments and editorial guidance helpful. Please do not hesitate to contact me if there is anything you would like to discuss. Thank you for considering NCB for your work,

Best wishes,

Melina

Melina Casadio, PhD
Senior Editor, Nature Cell Biology
ORCID ID: <https://orcid.org/0000-0003-2389-2243>

Reviewers' Comments:

Reviewer #1:

Remarks to the Author:

The Atg1/ULK kinase complex integrates various stresses to control autophagy initiation. Previous studies have shown that components and actions of the Atg1 complex are differentially explored in triggering autophagy under different stimulating conditions. For example, in glucose starvation-induced autophagy, Snf1-mediated phosphorylation of Mec1 leads to assembly of the Snf1-Mec1-Atg1 module on the mitochondrial surface that promotes the association of Atg1 with Atg13. A diverse class of factors interact with the adaptor subunit Atg11/FIP200 of the Atg1/ULK complex to modulate its activity, such as the association of TBK1 with FIP200 in xenophagy. In this manuscript, Gross and colleagues revealed that upon phosphate starvation, the metabolite sensor Pho81 interacts with Atg11 to modulate pexophagy. Interestingly, the core Atg1 kinase complex components Atg13 and Atg17 are dispensable for phosphate starvation-induced autophagy. Pho81 compensates for partially inactive Atg13 due to significant activity of TORC1 during phosphate starvation-triggered pexophagy. This study revealed that Pho81 integrates metabolic information directly to the Atg1 kinase complex for autophagy. Previous studies have already shown incorporation of energy/stress sensing factors in the Atg1 complex to modulate autophagy. This study does not provide sufficient novelty to advance our understanding of autophagosome initiation.

Major concerns:

1. This study solely based on the Om45-GFP and Pex11-GFP cleavage assays to monitor mitophagy and pexophagy, respectively. The GFP-cleavage assay is not robust, showing only mild impairment in Δ pho81 cells, and the cleavage may be affected by other irrelevant factors. Degradation of several endogenous mitochondrial or peroxisomal proteins should be determined. EM analysis should also be performed to directly visualize the selective engulfment of mitochondria and peroxisomes in autophagosomes.
2. The mechanism by which Pho81 activates the Atg1 complex should be addressed. Does Pho81 activate the Atg1 kinase activity or facilitate the recruitment of downstream Atg proteins?
3. During nitrogen starvation-induced autophagy, the multivalent interactions among Atg13 and Atg17 drive liquid-liquid phase separation of the Atg13-Atg17-Atg1 complex and the condensates are targeted to the vacuolar membrane via interacting with Vac8. Is the Pho81-Atg1 complex localized to the vacuolar membrane during phosphate starvation-induced autophagy? Atg13 and Atg17 are dispensable for phosphate starvation-induced autophagy. Does the Atg1-Pho81 complex form puncta on the vacuolar membrane? If yes, does Pho81 undergo LLPS?
4. The nature of the punctate structure labeled by Pho81-mCherry and GFP-Atg11 shown in Figure 1e and 1f should be identified. Does it correspond to the PAS? How does Pho81 trigger pexophagy? Are peroxisomes recruited to the Atg11-labeled PAS? At which stage is the pexophagy defective?
5. The authors evaluated bulk autophagy by measuring the cleavage of GFP-Atg8 assay. GFP-Atg8 is conjugated to autophagic membranes which could engulf selective or non-selective cargo. Other

assays should be performed to determine the role of Pho81 in bulk autophagy. In Figure 3e, the cleavage of 2GFP-Atg8 is enhanced in Δ pho81 and Δ pho81 Δ atg13 cells compared to the corresponding control. This suggests that pho81 deletion enhances the general autophagy activity. How to explain this contradictory result?

6. Page 9, the authors stated that Pho81 compensates for partially activated Atg13. The authors provided no evidence to show that Atg13 is partially activated in Δ pho81 cells. The phosphorylation level of Atg13 has not been quantified.

Minor concerns:

1. Figure 1D, the interaction between Pho81 and Atg11 under basal and phosphate starvation conditions should be run in the same gel for comparison. No Atg11 signal was detected in the input blots, which makes the comparison unreliable.
2. Figure 1e and 1f, Pho81-mCherry forms a large number of puncta and one of them is colocalized with GFP-Atg11. While in Pho81-mCherry overexpression cells, only one punctate structure is present. Please clarify the difference.
3. The authors demonstrated an involvement of SPX domain of Pho81 in pexophagy. However, the authors did not provide direct evidence to show that this domain is involved in sensing phospho-metabolites to regulate pexophagy.
4. Figure 1f, about 80% of Atg11 puncta are positive for Pho81. It may not be accurate to use Pho81-mCherry puncta formation as a proxy for Atg11 interaction as shown in Figure 4.

Reviewer #2:

Remarks to the Author:

In this manuscript from Gross et al, the authors explore how autophagy activation is regulated during phosphate starvation in yeast. Their objective is to describe how metabolite sensing contributes to both the activation of early autophagy biogenesis proteins and the choice of downstream potential selective cargoes. Focusing upon the Atg1 complex, they discover a reduced dependency upon the essential complex component Atg13 during phosphate starvation. They discover that the Atg1 complex also associates with a component of the phosphate sensing pathway, Pho81, that this interaction is mediated by Atg11, and that Pho81 colocalizes at accumulations of Atg11 in yeast. Through very nice biochemistry, they map the site of association to an FIR motif. During phosphate starvation, they also demonstrate an enhanced targeting of peroxisomes which depends on the presence of Atg11, and partly though incompletely on the availability of Pho81. Pexophagy is also only partly dependent upon Atg13, but is lost in a Pho81/Atg13 double knockout. From this result the authors conclude that these proteins functionally interact with Pho81 compensating for a "partially inactive" form of Atg13. They deem the role of Pho81 as a modulator of Atg1 activity, distinct from classic selective autophagy receptors, because it enables Atg1 activity despite persistent Torc activity in the cell.

Overall, the experiments are well done, the authors make a convincing case for Pho81 involvement in Atg1 complex activity, in binding Atg11, and in the selective targeting of peroxisomes during phosphate starvation. The discovery of a new composition of the Atg1 complex will be of general interest to the field.

However, the mechanism by which Pho81 works is less clear at this stage. The authors propose that Pho81 "compensates for partially inactive Atg13". Presumably there are two separate Atg1 complexes harboring either Pho81 or Atg13 though they are not otherwise described. Evidence that Pho81 "aids

in AKC clustering" is not demonstrated. Likewise, how the SPX domain coordinates regulation of Pho81 activity is not explored. A fuller understanding of the mechanism would elevate the authors model of how modulators regulate the AKC, though the paper in its present form is already very solid.

1) Do Atg13 and Pho81 function within the same complex? The authors envision Pho81 as a modulator of the otherwise canonical Atg1 complex, but it is striking that complete loss of Atg13 still allows for P-S dependent pexophagy as long as Pho81 is present. Fundamentally, what is the composition of the Atg1 complex supporting phosphate-starvation mediated pexophagy?

2) Is Atg13 phosphorylated during pexophagy? The authors show a single time course of Atg13 phosphorylation state (Fig 3F) which suggests a much less robust pattern of dephosphorylation than in comparable nutrient starvation, but nonetheless the Atg13 phosphorylation state appears to be dynamic over the first 8 hours. Given there is essentially no relationship between Pho81 expression and Atg8 turnover (Fig 3e), it is unlikely that very much of the autophagosome biogenesis occurring in the cell is associated with Pho81 activity and thus bulk Atg13 phosphorylation is maybe not the best way to detect Pho81 dependent events. Could the authors IP the Pho81 complex from WT and Δ Atg13 conditions, assess its composition and evaluate the phosphorylation state of the components?

Minor notes/questions

1) The authors describe a "drastic.." phosphate starvation dependent interaction of Pho81 and Atg11 (fig 1D). The ratio of mCherry to GFP changes dramatically, but the blot suggests rather comparable recovery of Pho81 and just less overall Atg11. Is this an accurate interpretation; do the levels of Atg11 go down during prolonged phosphate starvation?

2) In Fig 1, the authors show colocalization of proteins expressed off the endogenous locus and of proteins overexpressed (1e and 1f). the colocalization appears to be nearly 100%. Is there any colocalization in the absence of phosphate starvation? It is also not clear to which sets of experiments the quantification is connected. The figure legend is also a bit unclear, indicating the quantification as panel "g"...

3) Pg 7, line 162. The authors distinguish canonical SARs from the apparent activity of Pho81 on the evidence that Pho81 does not bind Atg8. In mammalian autophagy, there are clearly some cargo adaptors that function in an LC3-family independent manner (e.g. Shoemaker lab evidence for how NBR1 directly engages FIP200 through FIR; 2020 EMBO J.). I wonder, given the very limited dependence on Pho81 in the levels of GFP-Atg8 turnover during phosphate starvation, if a similar direct receptor-Atg11 interaction could be happening here. Does Pho81 interact with peroxisomes? Likewise, the authors state that Pho81 is positively regulating "Atg11-Atg36-driven pexophagy"; while this is likely the case based on previous literature, I do not think the necessity for Atg36 is formally tested here.

Reviewer #3:

Remarks to the Author:

This is an interesting paper by Gross et al. on the activation of selective autophagy pathway (pexophagy) in yeast during phosphate starvation (PS) in a manner that is different than nitrogen starvation (NS)-induced autophagy, thereby reflecting the flexibility of selective autophagy pathways depending on the metabolic environment. During autophagy, signals converge upon activation of the Atg1 kinase complex (AKC), which in yeast is comprised of the protein kinase Atg1, Atg13 and the

Atg17 complex (composed of Atg17, Atg29 and Atg31), and the selective autophagy adaptor, Atg11.

A. The key findings of this study are the following:

1. The authors show, as demonstrated previously (ref. 27), that during PS, bulk (measured by GFP-Atg8 degradation) autophagy is activated, but while this is completely dependent on the Atg13, Atg17/Atg29/Atg31 complex during NS, it is only reduced, but yet significantly active, in the absence of these key proteins of the core autophagy machinery, during PS. This suggested a dependence on Atg11, which was verified by the absence of autophagy in *atg11 atg13* and *atg11 atg17* cells during PS. The authors go on to show that the composition of the AKC is distinct during PS, from that shown to exist during NS in previous studies.
2. The Atg11-dependent autophagy depends on Pho81, a phospho-metabolite sensor and cdk (Pho80-Pho85 complex) inhibitor, which exists at the AKC during PS. Specifically, Pho81 associates and co-localizes with Atg1 at sites of autophagosome formation at the AKC during PS. Pho81 is only minimally required for autophagy during NS, but is required with Atg13 during PS.
3. Pho81 specifically promotes pexophagy, and to a much lesser extent, mitophagy, but the reason for this, as well as how Atg1 is activated, is unclear. Pho81 appears to functionally (physical interaction not addressed) interact with Atg13 in that during PS TORC1 is not completely inactivated, as it is during NS, and hence Atg13 is not fully dephosphorylated and activated. Under these conditions, Pho81 appears to play a role in activating Atg11- and Atg36-dependent pexophagy, but this appears not to require Atg13 itself (Fig. 4h, 5d).
4. Pho81 interacts with Atg11 using a motif that is similar to one described in other proteins for Atg11/FIP200 binding. However, it does not interact with Atg8, making it distinct from an SAR. Therefore Pho81 is referred to as an autophagy modulator and not a SAR. Additionally, an SPX domain at the N-terminus of Pho81 that senses IP6 and IP7 is required for pexophagy.
5. The authors propose that Atg11/FIP200 family proteins may bind SARs and modulators (like Pho81) during selective autophagy.

B. The manuscript is original and of general interest in terms of the canonical mechanisms of selective autophagy. It would be greatly improved by addressing the following questions.

C. In general, the data are of high quality. Loading controls are missing in most figures (e.g. Fig. 1 b, Fig. 2c, 3c-g). Label for Fig. 1g is missing.

Fig. 2 legend – change # to * in several places referring to p values.

D. Fine.

E. Conclusions are generally fine.

F. Improvements suggested

1. What is the physiological significance of pexophagy activation during phosphate starvation? How is this related to the sensing of IP6 and IP7? Does a pexophagy-defective strain (e.g. *atg36*) have any deficiency during phosphate starvation?

2. I was surprised to see a 50% level of bulk autophagy in *pho81 atg13* cells during PS (Fig. 3e). What is the explanation for this?

3. Previous work by Yokota et al. (ref. 27) had suggested that PS induced autophagy is weaker than that induced by NS because of slower inactivation of TORC1, and therefore slower activation of Atg13 by dephosphorylation. In view of this, what exactly does Pho81 association with Atg11 at the AKC do, both when Atg13 is present and when it is not? Specifically, how is Atg1 activated by Pho81 during PS and how is this kept in check when PS is not happening?

G, H. OK.

Methods should be written concisely, but should contain all elements necessary to allow interpretation and replication of the results. As a guideline, Methods sections typically do not exceed 3,000 words. The Methods should be divided into subsections listing reagents and techniques. When citing previous methods, accurate references should be provided and any alterations should be noted. Information must be provided about: antibody dilutions, company names, catalogue numbers and clone numbers for monoclonal antibodies; sequences of RNAi and cDNA probes/primers or company names and catalogue numbers if reagents are commercial; cell line names, sources and information on cell line identity and authentication. Animal studies and experiments involving human subjects must be reported in detail, identifying the committees approving the protocols. For studies involving human subjects/samples, a statement must be included confirming that informed consent was obtained. Statistical analyses and information on the reproducibility of experimental results should be provided in a section titled "Statistics and Reproducibility".

All Nature Cell Biology manuscripts submitted on or after March 21 2016 must include a Data availability statement as a separate section after Methods but before references, under the heading "Data Availability". For Springer Nature policies on data availability see <http://www.nature.com/authors/policies/availability.html>; for more information on this particular policy see <http://www.nature.com/authors/policies/data/data-availability-statements-data-citations.pdf>. The Data availability statement should include:

- Accession codes for primary datasets (generated during the study under consideration and

designated as "primary accessions") and secondary datasets (published datasets reanalysed during the study under consideration, designated as "referenced accessions"). For primary accessions data should be made public to coincide with publication of the manuscript. A list of data types for which submission to community-endorsed public repositories is mandated (including sequence, structure, microarray, deep sequencing data) can be found here <http://www.nature.com/authors/policies/availability.html#data>.

- Unique identifiers (accession codes, DOIs or other unique persistent identifier) and hyperlinks for datasets deposited in an approved repository, but for which data deposition is not mandated (see here for details <http://www.nature.com/sdata/data-policies/repositories>).
- At a minimum, please include a statement confirming that all relevant data are available from the authors, and/or are included with the manuscript (e.g. as source data or supplementary information), listing which data are included (e.g. by figure panels and data types) and mentioning any restrictions on availability.
- If a dataset has a Digital Object Identifier (DOI) as its unique identifier, we strongly encourage including this in the Reference list and citing the dataset in the Methods.

We recommend that you upload the step-by-step protocols used in this manuscript to the Protocol Exchange. More details can found at www.nature.com/protocolexchange/about.

All imaging data should be accompanied by scale bars, which should be defined in the legend. Cropped images of gels/blots are acceptable, but need to be accompanied by size markers, and to retain visible background signal within the linear range (i.e. should not be saturated). The boundaries of panels with low background have to be demarked with black lines. Splicing of panels should only be considered if unavoidable, and must be clearly marked on the figure, and noted in the legend with a statement on whether the samples were obtained and processed simultaneously. Quantitative comparisons between samples on different gels/blots are discouraged; if this is unavoidable, it should only be performed for samples derived from the same experiment with gels/blots were processed in parallel, which needs to be stated in the legend.

Figures should be provided at approximately the size that they are to be printed at (single column is 86 mm, double column is 170 mm) and should not exceed an A4 page (8.5 x 11"). Reduction to the scale that will be used on the page is not necessary, but multi-panel figures should be sized so that the whole figure can be reduced by the same amount at the smallest size at which essential details in each panel are visible. In the interest of our colour-blind readers we ask that you avoid using red and green for contrast in figures. Replacing red with magenta and green with turquoise are two possible

colour-safe alternatives. Lines with widths of less than 1 point should be avoided. Sans serif typefaces, such as Helvetica (preferred) or Arial should be used. All text that forms part of a figure should be rewritable and removable.

SUPPLEMENTARY INFORMATION – Supplementary information is material directly relevant to the conclusion of a paper, but which cannot be included in the printed version in order to keep the manuscript concise and accessible to the general reader. Supplementary information is an integral

part of a Nature Cell Biology publication and should be prepared and presented with as much care as the main display item, but it must not include non-essential data or text, which may be removed at the editor's discretion. All supplementary material is fully peer-reviewed and published online as part of the HTML version of the manuscript. Supplementary Figures and Supplementary Notes are appended at the end of the main PDF of the published manuscript.

The total number of Supplementary Figures (not including the "unprocessed scans" Supplementary Figure) should not exceed the number of main display items (figures and/or tables (see our Guide to Authors and March 2012 editorial <http://www.nature.com/ncb/authors/submit/index.html#suppinfo>; <http://www.nature.com/ncb/journal/v14/n3/index.html#ed>). No restrictions apply to Supplementary Tables or Videos, but we advise authors to be selective in including supplemental data.

GUIDELINES FOR EXPERIMENTAL AND STATISTICAL REPORTING

REPORTING REQUIREMENTS – We are trying to improve the quality of methods and statistics reporting in our papers. To that end, we are now asking authors to complete a reporting summary that collects information on experimental design and reagents. The Reporting Summary can be found here <https://www.nature.com/documents/nr-reporting-summary.pdf> If you would like to reference the guidance text as you complete the template, please access these flattened versions at <http://www.nature.com/authors/policies/availability.html>.

STATISTICS – Wherever statistics have been derived the legend needs to provide the n number (i.e. the sample size used to derive statistics) as a precise value (not a range), and define what this value represents. Error bars need to be defined in the legends (e.g. SD, SEM) together with a measure of centre (e.g. mean, median). Box plots need to be defined in terms of minima, maxima, centre, and percentiles. Ranges are more appropriate than standard errors for small data sets. Wherever statistical significance has been derived, precise p values need to be provided and the statistical test used needs to be stated in the legend. Statistics such as error bars must not be derived from $n < 3$. For

sample sizes of $n < 5$ please plot the individual data points rather than providing bar graphs. Deriving statistics from technical replicate samples, rather than biological replicates is strongly discouraged. Wherever statistical significance has been derived, precise p values need to be provided and the statistical test stated in the legend.

Author Rebuttal to Initial comments

We would like to thank the reviewers for their time and overall positive and very constructive comments and questions. We are excited about the newly added data, which provide significant insights into the molecular mechanisms underlying the function of the phospho-metabolite sensor Pho81 for phosphate starvation-induced pexophagy.

Reviewers' Comments:

Reviewer #1:

Remarks to the Author:

The Atg1/ULK kinase complex integrates various stresses to control autophagy initiation. Previous studies have shown that components and actions of the Atg1 complex are differentially explored in triggering autophagy under different stimulating conditions. For example, in glucose starvation-induced autophagy, Snf1-mediated phosphorylation of Mec1 leads to assembly of the Snf1-Mec1-Atg1 module on the mitochondrial surface that promotes the association of Atg1 with Atg13. A diverse class of factors interact with the adaptor subunit Atg11/FIP200 of the Atg1/ULK complex to modulate its activity, such as the

association of TBK1 with FIP200 in xenophagy. In this manuscript, Gross and colleagues revealed that upon phosphate starvation, the metabolite sensor Pho81 interacts with Atg11 to modulate pexophagy. Interestingly, the core Atg1 kinase complex components Atg13 and Atg17 are dispensable for phosphate starvation-induced autophagy. Pho81 compensates for partially inactive Atg13 due to significant activity of TORC1 during phosphate starvation-triggered pexophagy. This study revealed that Pho81 integrates metabolic information directly to the Atg1 kinase complex for autophagy. Previous studies have already shown incorporation of energy/stress sensing factors in the Atg1 complex to modulate autophagy. This study does not provide sufficient novelty to advance our understanding of autophagosome initiation.

Major concerns:

1. This study solely based on the Om45-GFP and Pex11-GFP cleavage assays to monitor mitophagy and pexophagy, respectively. The GFP-cleavage assay is not robust, showing only mild impairment in Δ pho81 cells, and the cleavage may be affected by other irrelevant factors. Degradation of several endogenous mitochondrial or peroxisomal proteins should be determined. EM analysis should also be performed to directly visualize the selective engulfment of mitochondria and peroxisomes in autophagosomes.

GFP-cleavage assays are a widely accepted standard to monitor turnover by autophagy in yeast in the field (Klionsky et al. 2021). It is true that the cleavage of GFP may be affected by known (and unknown) cellular processes. However, this is not unique to the GFP cleavage assay and can be said about any method the autophagy (or cell biology) field is using to assess substrate turnover. Nevertheless, we consider any factors, known or unknown, which influence substrate turnover, to inform us about underlying biology and thus relevant to our studies.

We use the term robust with regard to the reproducibility of our results. The effects of Pho81 on pexophagy are highly reproducible and thus robust. Clearly, the effects of a *PHO81* deletion on pexophagy alone are partial. However, this is a common observation in cell biology for factors that operate redundantly with other factors/pathways. We argue that the fact that cells have evolved redundancies argues for and not against the importance of these mechanisms. We view the functional redundancy of Pho81 and Atg13 for pexophagy during

phosphate starvation as a key observation in our study. Notably, in the absence of both, Pho81 and Atg13, pexophagy is fully inhibited.

We do share the reviewer's view to support key conclusions by independent methods. We have performed unbiased whole cell proteome analyses, which are fully consistent with Pho81-dependent mitophagy and pexophagy upon phosphate starvation (**new figures 2a and b**). In addition, we monitored changes in abundance of peroxisomes by cytological analysis upon phosphate starvation, which fully support both, our Pex11-GFP turnover and proteomics data (**Figure 2f**). Thus, we provide three independent lines of evidence for the involvement of Pho81 in the regulation of pexophagy in our study.

The principal mechanisms of pexophagy have been established in molecular detail by a number of groups including EM analyses to demonstrate the presence of peroxisomes within autophagosomes. We have now shown that cells use the same canonical Atg11-Atg36-Pex3 protein machinery for pexophagy for both, nitrogen and phosphate starvation (**new figures 2d, e**). In our study, only a small fraction of autophagosomes is expected to engage in pexophagy during phosphate starvation-induced autophagy. Thus, it would be very labor and time intensive to identify autophagosomes, which contain peroxisomes. More importantly, it is virtually impossible to perform the required quantitative analyses by EM to add to our work interrogating the function of Pho81 and Atg13 for pexophagy during phosphate starvation. Thus, we think EM analyses would not or is ill-suited to sufficiently add to our understanding of the Pho81-dependent mechanisms of pexophagy to justify the associated time and effort.

2. The mechanism by which Pho81 activates the Atg1 complex should be addressed. Does Pho81 activate the Atg1 kinase activity or facilitate the recruitment of downstream Atg proteins?

We absolutely agree that this is a key question in our study. Our revised work has now uncovered two molecular mechanisms, which define the function of Pho81 at the AKC for pexophagy. First, we provide whole cell phosphoproteomics, genetic, and pharmacological evidence that (a) TORC1 retains significant activity during phosphate starvation and (b) the level of Atg13 phosphorylation controlled by TORC1 is a key factor, which determines whether cells require the function of Pho81 at the AKC for pexophagy (**new Figs. 3i-k**). Second, we performed phosphoproteomics of AKC subunits after co-immunopurification of GFP-Atg11

and MS analysis. Excitingly, we have identified three phosphorylation sites in Atg11, S121, S613 and S935, which show a significant reduction in phosphorylation when Pho81 is unable to bind to the AKC via Atg11 (**new Fig. 4i**). S613 and S935 in Atg11 have previously been identified as target sites of the Atg1 kinase (Hu et al. 2019). We now show that the majority of Pho81-dependent pexophagy is significantly impaired in the presence of an unphosphorylatable *atg11^{3SA}* variant mutated for all three phosphosites, in a manner that is epistatic with the absence of Pho81 (**new Fig. 4j**). These data strongly indicate that the main function of Pho81 is to promote Atg11 phosphorylation very likely by Atg1 to promote pexophagy. Taken together, our work provides first evidence for a conditional subunit of the AKC, which regulates pexophagy by binding to Atg11 and promoting the phosphorylation of Atg11 by Atg1. Thus, our work provides substantial insights into a new regulatory mechanism of Pho81 at the AKC to promote pexophagy during phosphate starvation in support of our proposed model.

3. During nitrogen starvation-induced autophagy, the multivalent interactions among Atg13 and Atg17 drive liquid-liquid phase separation of the Atg13-Atg17-Atg1 complex and the condensates are targeted to the vacuolar membrane via interacting with Vac8. Is the Pho81-Atg1 complex localized to the vacuolar membrane during phosphate starvation-induced autophagy? Atg13 and Atg17 are dispensable for phosphate starvation-induced autophagy. Does the Atg1-Pho81 complex form puncta on the vacuolar membrane? If yes, does Pho81 undergo LLPS?

We have now analyzed the functional relationship of Pho81 and Atg17 (**new fig. 3c**) and find that Atg17, similar to Atg13, functionally interacts with Pho81 in pexophagy during phosphate starvation. These data now show that Atg13-Atg17-dependent interactions provide a partially redundant function with Pho81 and Atg11. Upon phosphate starvation, endogenously expressed Pho81-GFP localizes to FM4-64- and Vac8-mCherry-positive structures (**Reviewer Fig. 2a, b**). Moreover, endogenously co-expressed Pho81-GFP and Atg1-mCherry co-localize in vicinity of CMAC-stained vacuoles (**Reviewer Fig. 2c**). Intriguingly, loss of *ATG13* strongly increases the number of Atg1-Pho81-positive puncta at the vacuole (**Reviewer Fig. 2c**). Taken together, these data show that Pho81-positive AKCs form punctate structures at vacuolar membranes during phosphate starvation.

While LLPS is a critical feature of AKC-driven induction of bulk autophagy, a role for LLPS in selective forms of autophagy is much less clear. In fact, work from the Denic, Kraft, and Joule and other labs suggest that AKC crowding upon receptor binding on cargo substrate surfaces likely is the driving force for AKC activation. We think that AKC clustering by Atg13-17 and Pho81-Atg11-dependent interactions is a likely model. However, whether these interactions result in physical changes that qualify as LLPS might be a secondary question. Thus, while it is an interesting question to explore possible contributions of LLPS-driven mechanisms for Pho81-dependent pexophagy, we consider this question beyond the scope of our current study.

Reviewer Fig. 2: Fluorescence imaging of, **a**, genomic Pho81-GFP and Vac8-mCherry after phosphate starvation (4 h); of, **b**, genomic Pho81-GFP in WT or Δ atg13 cells stained with the vital dye, FM4-64, for vacuolar membranes after phosphate starvation (4 h); and of, **c**, genomic Pho81-GFP and Atg1-mCherry with vacuolar lumen stained with CellTracker™ CMAC.

4. The nature of the punctate structure labeled by Pho81-mCherry and GFP-Atg11 shown in Figure 1e and 1f should be identified. Does it correspond to the PAS? How does Pho81 trigger pexophagy? Are peroxisomes recruited to the Atg11-labeled PAS? At which stage is the pexophagy defective?

Our data show that Pho81 localizes to Atg1-, Atg8-, Atg11-, and Atg13-marked puncta, key markers for a PAS. Moreover, Pho81 localizes to Atg1 puncta in a manner strongly enhanced in the absence of Atg13 (**new extended data 1**), indicating competitive interactions between Pho81 and Atg13 at Atg1-marked structures. Additionally, Pho81-positive AKCs form punctate structures at vacuolar membranes during phosphate starvation (s. above and **reviewer figure 2**). Taken together, these data strongly suggest that Pho81 is recruited to PASs.

Given that under our experimental conditions, Pho81-Atg11-dependent pexophagy constitutes a minor fraction of autophagosome biogenesis events and Pho81 does not affect bulk autophagy mechanisms, the question of which downstream factors might be affected in the absence of Pho81 is challenging to address and at this point we lack the tools to do so. However, as outlined above, our data clearly place the function of Pho81 at the AKC and Atg11 regulation.

5. The authors evaluated bulk autophagy by measuring the cleavage of GFP-Atg8 assay. GFP-Atg8 is conjugated to autophagic membranes which could engulf selective or non-selective cargo. Other assays should be performed to determine the role of Pho81 in bulk autophagy. In Figure 3e, the cleavage of 2GFP-Atg8 is enhanced in Δ pho81 and Δ pho81 Δ atg13 cells compared to the corresponding control. This suggests that pho81 deletion enhances the general autophagy activity. How to explain this contradictory result?

In addition to 2GFP-Atg8, we now have analyzed the turnover of cytosolic tandem GFP-GFP (2GFP) (**new figure 3f**) and, fully consistent with our conclusions, found that bulk autophagy strictly depends on Atg13 independent of Pho81. While we did observe a tendency for slightly increased 2GFP-Atg8 (or 2GFP) turnover in the absence of Pho81, it did not reach statistical significance, even when we included additional independent replicates (n=10)(**Figure 3e, f**). However, even if true, we do not consider this a contradiction to our model. First, cells may upregulate bulk or other selective forms of autophagy in the absence of Pho81-dependent pexophagy. Second, we cannot exclude that the binding of Pho81 to Atg11 might not only promote pexophagy but also mildly reduce the induction of bulk or other selective forms of autophagy.

6. Page 9, the authors stated that Pho81 compensates for partially activated Atg13. The authors provided no evidence to show that Atg13 is partially activated in Δ pho81 cells. The phosphorylation level of Atg13 has not been quantified.

We have based our conclusion on two lines of evidence. First, global phosphoproteomics and Atg13 phosphorylation patterns show that TORC1 retains significant activity during phosphate starvation (**new figures 3g-l**). Second, we now show that Atg13 phosphorylation is not affected by the absence of Pho81 (**new figure 3j**), but pexophagy is rendered fully Pho81-independent when TORC1 is inhibited and, as a consequence, Atg13 is fully dephosphorylated in the presence of rapamycin during phosphate starvation (**new figure 3k**). Moreover, in the absence of the kinase Npr2, which is required for TORC1 inhibition, pexophagy largely depends on the presence of Pho81 (**new figure 3l**). Taken together, these data strongly support the conclusion that the level of Atg13 phosphorylation determines the Pho81-dependency of pexophagy during phosphate starvation.

Minor concerns:

1. Figure 1D, the interaction between Pho81 and Atg11 under basal and phosphate starvation conditions should be run in the same gel for comparison. No Atg11 signal was detected in the input blots, which makes the comparison unreliable.

The samples for growing conditions and phosphate starvation were run in parallel on the same gel and Western blots (full scan in source data). However, we ran the samples twice on separate gels in order to detect Pho81-mCherry and GFP-Atg11 independently, which run at the approximately same mass in SDS-PAGE. Thus, this experimental setup allows us to determine the relative changes in Pho81 and Atg11 interactions. Because we normalize the signals of bound Pho81-mCherry to the signals of immunisolated GFP-Atg11, the results are not affected by fact that we did not detect GFP-Atg11 in our input controls.

2. Figure 1e and 1f, Pho81-mCherry forms a large number of puncta and one of them is colocalized with GFP-Atg11. While in Pho81-mCherry overexpression cells, only one punctate structure is present. Please clarify the difference.

We consistently observe cortical or peripheral punctate structures for genomic Pho81-mCherry (**Figures 1e, 2g, and 5e**), which is fully consistent with very recently published work (Chabert et al. eLife 2023). Overexpression of Pho81 likely results in higher cytosolic levels, which may mask cortical structures.

3. The authors demonstrated an involvement of SPX domain of Pho81 in pexophagy. However, the authors did not provide direct evidence to show that this domain is involved in sensing phospho-metabolites to regulate pexophagy.

In the revised manuscript, we have added substantial new data to explore the role of phospho-metabolite-sensing by the SPX domain for Pho81-dependent pexophagy. Our new data indicate that phospho-metabolite binding to the SPX-domain negatively regulates Pho81 localization to Atg1-marked puncta and formation of cortical Pho81 structures (**new figure 5e**). Interestingly, the SPX domain itself is required for cortical localization of Pho81 (**new figure 5e**). In addition, we find that uncontrolled excess binding of Pho81 to the AKC can inhibit pexophagy or generally selective autophagy (**new figures 5f-h**). To assess the role of InsP-regulation of autophagy, we supplemented InsP6 during phosphate starvation and observed a partially Pho81-dependent inhibition of pexophagy (**new figure 5i**). Moreover, overexpression of the inositol polyphosphate kinase Kcs1 also reduced pexophagy in

phosphate-starved cells (**new figure 5j**). Taken together, these data show that (1) the SPX domain regulates Pho81 binding to Atg11/AKC and (2) phospho-metabolites regulate pexophagy/autophagy during phosphate starvation.

4. Figure 1f, about 80% of Atg11 puncta are positive for Pho81. It may not be accurate to use Pho81-mCherry puncta formation as a proxy for Atg11 interaction as shown in Figure 4.

100% of overexpressed Pho81-mCherry puncta colocalized with GFP-Atg11, suggesting that Pho81 puncta formation serves as a reasonable first readout for Atg11 interaction, albeit in an indirect manner. Importantly, all of our data were independently confirmed by Y2H analysis of Pho81-Atg11 interactions (**Figure 4b**).

Reviewer #2:

Remarks to the Author:

In this manuscript from Gross et al, the authors explore how autophagy activation is regulated during phosphate starvation in yeast. Their objective is to describe how metabolite sensing contributes to both the activation of early autophagy biogenesis proteins and the choice of downstream potential selective cargoes. Focusing upon the Atg1 complex, they discover a reduced dependency upon the essential complex component Atg13 during phosphate starvation. They discover that the Atg1 complex also associates with a component of the phosphate sensing pathway, Pho81, that this interaction is mediated by Atg11, and that Pho81 colocalizes at accumulations of Atg11 in yeast. Through very nice biochemistry, they map the site of association to an FIR motif. During phosphate starvation, they also demonstrate an enhanced targeting of peroxisomes which depends on the presence of Atg11, and partly though incompletely on the availability of Pho81. Pexophagy is also only partly dependent upon Atg13, but is lost in a Pho81/Atg13 double knockout. From this result the authors conclude that these proteins functionally interact with Pho81 compensating for a "partially inactive" form of Atg13. They deem the role of Pho81 as a modulator of Atg1 activity, distinct from classic selective autophagy receptors, because it enables Atg1 activity despite persistent Torc activity in the cell.

Overall, the experiments are well done, the authors make a convincing case for Pho81 involvement in Atg1 complex activity, in binding Atg11, and in the selective targeting of

peroxisomes during phosphate starvation. The discovery of a new composition of the Atg1 complex will be of general interest to the field.

However, the mechanism by which Pho81 works is less clear at this stage. The authors propose that Pho81 “compensates for partially inactive Atg13”. Presumably there are two separate Atg1 complexes harboring either Pho81 or Atg13 though they are not otherwise described. Evidence that Pho81 “aids in AKC clustering” is not demonstrated.

We absolutely agree that this is a key question in our study. Our revised work has now uncovered two molecular mechanisms, which define the function of Pho81 at the AKC for pexophagy. First, we provide whole cell phosphoproteomics, genetic, and pharmacological evidence that (a) TORC1 retains significant activity during phosphate starvation and (b) the level of Atg13 phosphorylation controlled by TORC1 is a key factor, which determines whether cells require the function of Pho81 at the AKC for pexophagy (**new Figs. 3i-k**). Second, we performed phosphoproteomics of AKC subunits after co-immunopurification of GFP-Atg11 and MS analysis. Excitingly, we have identified three phosphorylation sites in Atg11, S121, S613 and S935, which show a significant reduction in phosphorylation when Pho81 is unable to bind to the AKC via Atg11 (**new Fig. 4i**). S613 and S935 in Atg11 have previously been identified as target sites of the Atg1 kinase (Hu et al. 2019). We now show that the majority of Pho81-dependent pexophagy is significantly impaired in the presence of an unphosphorylatable *atg11^{3SA}* variant mutated for all three phosphosites, in a manner that is epistatic with the absence of Pho81 (**new Fig. 4j**). Taken together, our work provides first evidence for a conditional subunit of the AKC, which regulates pexophagy by binding to Atg11 and promoting the phosphorylation of Atg11 very likely by Atg1. Thus, our work provides substantial insights into a new regulatory mechanism of Pho81 at the AKC to promote pexophagy during phosphate starvation in support of our proposed model.

Likewise, how the SPX domain coordinates regulation of Pho81 activity is not explored. A fuller understanding of the mechanism would elevate the authors model of how modulators regulate the AKC, though the paper in its present form is already very solid.

In the revised manuscript, we have added substantial new data to explore the role of phospho-metabolite-sensing by the SPX domain for Pho81-dependent pexophagy. Our new data indicate that phospho-metabolite binding to the SPX-domain negatively regulates Pho81 localization to Atg1-marked puncta and formation of cortical Pho81 structures (**new figure 5e**). Interestingly, the SPX domain itself is required for cortical localization of Pho81 (**new figure 5e**). In addition, we find that uncontrolled excess binding of Pho81 to the AKC can inhibit pexophagy or generally selective autophagy (**new figures 5f-h**). To assess the role of InsP-regulation of autophagy, we supplemented InsP6 during phosphate starvation and observed a partially Pho81-dependent inhibition of pexophagy (**new figure 5i**). Moreover, overexpression of the inositol polyphosphate kinase Kcs1 also reduced pexophagy in phosphate-starved cells (**new figure 5j**). Taken together, these data show that (1) the SPX domain regulates Pho81 binding to Atg11/AKC and (2) phospho-metabolites regulate pexophagy/autophagy during phosphate starvation.

1) Do Atg13 and Pho81 function within the same complex? The authors envision Pho81 as a modulator of the otherwise canonical Atg1 complex, but it is striking that complete loss of Atg13 still allows for P-S dependent pexophagy as long as Pho81 is present. Fundamentally, what is the composition of the Atg1 complex supporting phosphate-starvation mediated pexophagy?

Performing fluorescence imaging, we found that Pho81-GFP and Atg13-mCherry localized to the same PAS during phosphate starvation (**new extended data 1c**). Moreover, Pho81-GFP localizes to Atg1-mCherry-marked puncta, which significantly increases in the absence of Atg13 (**new extended data 1d**). These data suggest a, at least partially, competitive association of Pho81 and Atg13 with Atg1. To determine the AKC composition for selective autophagy, we performed immunopurification followed by MS-analysis of GFP-Atg11-interacting proteins and found that both Pho81 and Atg13 are bound to Atg11-containing AKCs, however, Atg13 enrichment was often lower compared with other AKC subunits (**Figure 5c**). Thus, at this point we cannot conclusively distinguish whether Atg13 and Pho81 operate in clearly separate, but functionally redundant complexes, within the same AKC assembly, or a mixture of the two possibilities. We currently do not have the genetic or biochemical tools

to isolate and characterize only AKCs engaged in pexophagy, since phosphate starvation elicits a broad autophagy response composed of bulk and selective forms, which do not always depend on Pho81. However, our new data clearly show that Pho81 compensates for the absence of both, Atg13 and Atg17 (**new figure 3b, c**), indicating that Pho81 critically drives Atg1-Atg11-mediated pexophagy. Mechanistically, as discussed, we show that Pho81-binding to Atg11 is required for Atg11 phosphorylation and pexophagy. We are highly interested in exploring the functional consequences of Atg11 phosphorylation for the induction of pexophagy, which likely will shed light on the functional interaction of Pho81 with Atg13. However, this will require a substantial amount of in vitro biochemical and structural work. While we are working towards these analyses, we consider these experiments to be beyond the scope of the current manuscript.

2) Is Atg13 phosphorylated during pexophagy? The authors show a single time course of Atg13 phosphorylation state (Fig 3F) which suggests a much less robust pattern of dephosphorylation than in comparable nutrient starvation, but nonetheless the Atg13 phosphorylation state appears to be dynamic over the first 8 hours. Given there is essentially no relationship between Pho81 expression and Atg8 turnover (Fig 3e), it is unlikely that very much of the autophagosome biogenesis occurring in the cell is associated with Pho81 activity and thus bulk Atg13 phosphorylation is maybe not the best way to detect Pho81 dependent events. Could the authors IP the Pho81 complex from WT and deltaAtg13 conditions, assess its composition and evaluate the phosphorylation state of the components?

Our pharmacological and genetic data indicate that TORC1-dependent phosphorylation of Atg13 determines whether pexophagy depends on the presence of Pho81 during phosphate starvation (**new Figures 3j-l**). Specifically, we show that rapamycin treatment during phosphate starvation results in strong dephosphorylation of Atg13 and in Pho81-independent pexophagy. In turn, in the absence of the TORC1-inhibitor kinase Npr2, pexophagy is largely dependent on Pho81 (**Figures 3j-k**). Moreover, we performed whole cell phosphoproteomics and found a clear signature of elevated TORC1 activity in phosphate starved cells compared with nitrogen starvation (**new figures 3g-i**). Thus, we propose that Pho81 becomes relevant in the presence of significant TORC1 activity and limited Atg13 dephosphorylation to promote pexophagy. How the differences in phosphorylation of Atg13

during nitrogen and phosphate starvation affect the regulation of Pho81-independent bulk autophagy is an area we are very interested in exploring in the future. Our phosphoproteomics data suggest a complex Atg1 phosphosite pattern for autophagy proteins during phosphate starvation compared with nitrogen starvation (**new figure 3i**), suggesting that we are only beginning to explore the substantial plasticity in phosphoregulation of the autophagy protein machinery. However, these mechanisms clearly occur in a Pho81-independent manner and, thus, are not the scope of this manuscript.

Minor notes/questions

1) The authors describe a "drastic.." phosphate starvation dependent interaction of Pho81 and Atg11 (fig 1D). The ratio of mCherry to GFP changes dramatically, but the blot suggests rather comparable recovery of Pho81 and just less overall Atg11. Is this an accurate interpretation; do the levels of Atg11 go down during prolonged phosphate starvation?

We performed GFP-Atg11 immuno-purifications to assess the binding of Pho81. We normalized the data to the protein level of purified Atg11 to calculate the ratio of Pho81/Atg11 during growth or after phosphate starvation. We do observe a decrease in Atg11, likely due to degradation by autophagy. However, the relative association of Pho81 with Atg11 in our data is measured in a way that is independent of the absolute steady-state level of Atg11.

2) In Fig 1, the authors show colocalization of proteins expressed off the endogenous locus and of proteins overexpressed (1e and 1f). the colocalization appears to be nearly 100%. Is there any colocalization in the absence of phosphate starvation? It is also not clear to which sets of experiments the quantification is connected. The figure legend is also a bit unclear, indicating the quantification as panel "g"...

Upon overexpression, Pho81 can constitutively associates with Atg11 (**data not shown**) similar to Pho81 variants without function SPX domains (**new figure 5e**). In fact, overexpression of Pho81 results in uncontrolled excess binding to Atg11, which inhibits pexophagy or general selective autophagy in the absence of Atg13 (**new figures 5g, h**)

3) Pg 7, line 162. The authors distinguish canonical SARs from the apparent activity of Pho81 on the evidence that Pho81 does not bind Atg8. In mammalian autophagy, there are clearly some cargo adaptors that function in an LC3-family independent manner (e.g. Shoemaker lab evidence for how NBR1 directly engages FIP200 through FIR; 2020 EMBO J.). I wonder, given the very limited dependence on Pho81 in the levels of GFP-Atg8 turnover during phosphate starvation, if a similar direct receptor-Atg11 interaction could be happening here. Does Pho81 interact with peroxisomes?

It is interesting to note that the Nbr1 receptor contains a LIR motif and binds to Atg8 proteins (REF). The Shoemaker lab beautifully characterized Atg8 protein-independent turnover of Nbr1 itself, but did not assess the degradation of Nbr1 cargos (aggrephagy or pexophagy). However, the Lazarou lab has shown that Atg8 proteins are dispensable for mitophagy in HeLa cells (Nguyen et al. JCB 2016), suggesting that receptor-Atg8 interactions are not strictly required for selective autophagy, but it seems to be part of the mode of action to ensure "exclusive" cargo degradation. To further test whether Pho81 might function as an Atg8-independent receptor, we examined its cellular localization. Interestingly, Pho81 formed cortical punctate structures often found in close proximity of peripheral peroxisomes independent of Atg11 or Atg36 (**new figure 2g**). However, it did not mark the surface of peroxisomes in contrast to the canonical pexophagy receptor Atg36 or co-receptor Pex3 (Motley et al. EMBOJ 2012). In addition, the canonical pexophagy machinery of Atg11, Atg36, and Pex3 is essential for pexophagy, while Pho81 functionally cooperates with Atg13/Atg17 to promote pexophagy. Based on our discovery that Pho81 regulates Atg11 phosphorylation to promote pexophagy, we interpret our data that Pho81 does not act as a canonical SAR and rather promotes pexophagy by regulating Atg11 within the AKC.

Likewise, the authors state that Pho81 is positively regulating “Atg11-Atg36-driven pexophagy”; while this is likely the case based on previous literature, I do not think the necessity for Atg36 is formally tested here.

We showed in the previous version of the manuscript that pexophagy depends on Atg11 and Atg36 during phosphate starvation (**Figure 2d**). We now include data demonstrating that phosphate starvation-induced pexophagy also requires the co-receptor Pex3 (**new figure 2e**), indicating that cells use the same canonical Atg11-Atg36-Pex3 protein machinery for pexophagy during both, phosphate and nitrogen starvation.

Reviewer #3:

Remarks to the Author:

This is an interesting paper by Gross et al. on the activation of selective autophagy pathway (pexophagy) in yeast during phosphate starvation (PS) in a manner that is different than nitrogen starvation (NS)-induced autophagy, thereby reflecting the flexibility of selective autophagy pathways depending on the metabolic environment. During autophagy, signals converge upon activation of the Atg1 kinase complex (AKC), which in yeast is comprised of the protein kinase Atg1, Atg13 and the Atg17 complex (composed of Atg17, Atg29 and Atg31), and the selective autophagy adaptor, Atg11.

A. The key findings of this study are the following:

1. The authors show, as demonstrated previously (ref. 27), that during PS, bulk (measured by GFP-Atg8 degradation) autophagy is activated, but while this is completely dependent on the Atg13, Atg17/Atg29/Atg31 complex during NS, it is only reduced, but yet significantly active, in the absence of these key proteins of the core autophagy machinery, during PS. This suggested a dependence on Atg11, which was verified by the absence of autophagy in *atg11 atg13* and *atg11 atg17* cells during PS. The authors go on to show that the composition of the AKC is distinct during PS, from that shown to exist during NS in previous studies.
2. The Atg11-dependent autophagy depends on Pho81, a phospho-metabolite sensor and

cdk (Pho80-Pho85 complex) inhibitor, which exists at the AKC during PS. Specifically, Pho81 associates and co-localizes with Atg1 at sites of autophagosome formation at the AKC during PS. Pho81 is only minimally required for autophagy during NS, but is required with Atg13 during PS.

3. Pho81 specifically promotes pexophagy, and to a much lesser extent, mitophagy, but the reason for this, as well as how Atg1 is activated, is unclear. Pho81 appears to functionally (physical interaction not addressed) interact with Atg13 in that during PS TORC1 is not completely inactivated, as it is during NS, and hence Atg13 is not fully dephosphorylated and activated. Under these conditions, Pho81 appears to play a role in activating Atg11- and Atg36-dependent pexophagy, but this appears not to require Atg13 itself (Fig. 4h, 5d).

4. Pho81 interacts with Atg11 using a motif that is similar to one described in other proteins for Atg11/FIP200 binding. However, it does not interact with Atg8, making it distinct from an SAR. Therefore Pho81 is referred to as an autophagy modulator and not a SAR.

Additionally, an SPX domain at the N-terminus of Pho81 that senses IP6 and IP7 is required for pexophagy.

5. The authors propose that Atg11/FIP200 family proteins may bind SARs and modulators (like Pho81) during selective autophagy.

B. The manuscript is original and of general interest in terms of the canonical mechanisms of selective autophagy. It would be greatly improved by addressing the following questions.

C. In general, the data are of high quality. Loading controls are missing in most figures (e.g. Fig. 1 b, Fig. 2c, 3c-g). Label for Fig. 1g is missing.

We measure the relative ratio of free GFP against the total GFP signal (free GFP + full length protein-GFP). The ratio is independent of loading and, therefore, does not require a loading control. We corrected the legend for figure 1.

1. What is the physiological significance of pexophagy activation during phosphate starvation? Does a pexophagy-defective strain (e.g. *atg36*) have any deficiency during phosphate starvation?

Our work indicates that Pho81-dependent pexophagy regulates the abundance of peroxisomes during phosphate starvation. We are in the process of profiling potential consequences for autophagy-dependent lipidome and proteome remodeling in cells in response to phosphate starvation. However, these datasets are complex and will require substantial experimentation to dissect.

2. I was surprised to see a 50% level of bulk autophagy in *pho81 atg13* cells during PS (Fig. 3e). What is the explanation for this?

We observed a general 50% reduction in 2GFP-Atg8 turnover in the absence of Atg13 during phosphate starvation (**Figures 1b, 3e**). However, bulk autophagy of cytosol is fully blocked in $\Delta atg13$ cells, when we measure the degradation of cytosolic tandem 2GFP (**new figure 3f**). Since 2GFP-Atg8 turnover is fully blocked in $\Delta atg11\Delta atg13$ cells, we conclude that the 50% remaining turnover in $\Delta atg13$ cells is caused by Atg11-mediated selective autophagy including pexophagy, mitophagy, and potentially other forms (**figure 2c, d**)

3. Previous work by Yokota et al. (ref. 27) had suggested that PS induced autophagy is weaker than that induced by NS because of slower inactivation of TORC1, and therefore slower activation of Atg13 by dephosphorylation. In view of this, what exactly does Pho81 association with Atg11 at the AKC do, both when Atg13 is present and when it is not?

We absolutely agree that this is a key question in our study. Our revised work has now uncovered two molecular mechanisms, which define the function of Pho81 at the AKC for pexophagy. First, we provide whole cell phosphoproteomics, genetic, and pharmacological evidence that (a) TORC1 retains significant activity during phosphate starvation and (b) the level of Atg13 phosphorylation controlled by TORC1 is a key factor, which determines whether cells require the function of Pho81 at the AKC for pexophagy (**new Figs. 3i-k**). Second, we performed phosphoproteomics of AKC subunits after co-immunopurification of GFP-Atg11 and MS analysis. Excitingly, we have identified three phosphorylation sites in Atg11, S121, S613 and S935, which show a significant reduction in phosphorylation when Pho81 is unable to bind to the AKC via Atg11 (**new Fig. 4i**). S613 and S935 in Atg11 have previously been identified as target sites of the Atg1 kinase (Hu et al. 2019). We now show that the majority

of Pho81-dependent pexophagy is significantly impaired in the presence of an unphosphorylatable *atg11^{3SA}* variant mutated for all three phosphosites, in a manner that is epistatic with the absence of Pho81 (**new Fig. 4 j**). Taken together, our work provides first evidence for a conditional subunit of the AKC, which regulates pexophagy by binding to Atg11 and promoting the phosphorylation of Atg11 very likely by Atg1. Thus, our work provides substantial insights into a new regulatory mechanism of Pho81 at the AKC to promote pexophagy during phosphate starvation in support of our proposed model.

Specifically, how is Atg1 activated by Pho81 during PS and how is this kept in check when PS is not happening? How is this related to the sensing of IP6 and IP7?

In the revised manuscript, we have added substantial new data to further support the role of phospho-metabolite-sensing by the SPX domain for Pho81-dependent pexophagy. Our new data indicate that phospho-metabolite binding to the SPX-domain negatively regulates Pho81 localization to Atg1-marked puncta and formation of cortical Pho81 structures (**new figure 5e**). Interestingly, the SPX domain itself is required for cortical localization of Pho81 (**new figure 5e**). In addition, we find that uncontrolled excess binding of Pho81 to the AKC can inhibit pexophagy or generally selective autophagy (**new figures 5f-h**). To assess the role of InsP-regulation of autophagy, we supplemented InsP6 during phosphate starvation and observed a partially Pho81-dependent inhibition of pexophagy (**new figure 5i**). Moreover, overexpression of the inositol polyphosphate kinase Kcs1 also reduced pexophagy in phosphate-starved cells (**new figure 5j**). Taken together, these data show that (1) the SPX domain regulates Pho81 binding to Atg11/AKC and (2) phospho-metabolites regulate pexophagy/autophagy during phosphate starvation.

Decision Letter, first revision:

27th November 2023

Dear Dr. Graef,

Thank you for submitting your revised manuscript "A metabolite sensor subunit of the Atg1/ULK

complex regulates selective autophagy" (NCB-LE50175A). It has now been seen by the original referees and their comments are below. The reviewers find that the paper has improved in revision, and therefore we'll be happy in principle to publish it in Nature Cell Biology, pending minor revisions to satisfy the referees' final requests and to comply with our editorial and formatting guidelines.

We are now performing detailed checks on your paper and will send you a checklist detailing our editorial and formatting requirements in about 1-2 weeks. Please do not upload the final materials and make any revisions until you receive this additional information from us.

Thank you again for your interest in Nature Cell Biology and for all your efforts in revision. Please do not hesitate to contact me if you have any questions.

Sincerely,

Melina

Melina Casadio, PhD
Senior Editor, Nature Cell Biology
ORCID ID: <https://orcid.org/0000-0003-2389-2243>

Reviewer #1 (Remarks to the Author):

The authors have performed sufficient experiments to address my concerns.

Reviewer #2 (Remarks to the Author):

In this revised submission, the authors have largely addressed my major concerns.

In particular, the evidence that Pho81 can partially compensate for Atg13/Atg17 in pexophagy is convincing. The authors also explore how phosphorylation governs the function of the putative Pho81-AKC complex through phosphoproteomics under conditions where WT or a mutant Pho81 is present. They discover three phosphorites on Atg11 are dependent upon WT Pho81 (thought to be the form engaging the AKC) and demonstrate these sites are necessary to support efficient pexophagy.

The new data exploring the consequences of the SPX domain demonstrates a clear role for this motif in intracellular localization including targeting to Atg1 puncta. Experiments intended to test a role for IP6 in regulating this interaction are consistent with the apparent negative regulation observed in imaging, though these experiments could be more fleshed out.

In total, this is a significant improvement on the already solid initial submission and will be of broad interest to the field.

Reviewer #3 (Remarks to the Author):

My comments have been adequately addressed in this revision.

I would suggest the explicit addition to the Fig 1 legend or Materials and methods that they measure the relative ratio of free GFP to total GFP

Decision Letter, final checks:

Our ref: NCB-LE50175A

6th December 2023

Dear Dr. Graef,

Thank you for your patience as we've prepared the guidelines for final submission of your Nature Cell Biology manuscript, "A metabolite sensor subunit of the Atg1/ULK complex regulates selective autophagy" (NCB-LE50175A). Please carefully follow the step-by-step instructions provided in the attached file, and add a response in each row of the table to indicate the changes that you have made. Please also check and comment on any additional marked-up edits we have proposed within the text. Ensuring that each point is addressed will help to ensure that your revised manuscript can be swiftly handed over to our production team.

In recognition of the time and expertise our reviewers provide to Nature Cell Biology's editorial process, we would like to formally acknowledge their contribution to the external peer review of your manuscript entitled "A metabolite sensor subunit of the Atg1/ULK complex regulates selective autophagy". For those reviewers who give their assent, we will be publishing their names alongside the published article.

Nature Cell Biology offers a Transparent Peer Review option for new original research manuscripts submitted after December 1st, 2019. As part of this initiative, we encourage our authors to support increased transparency into the peer review process by agreeing to have the reviewer comments, author rebuttal letters, and editorial decision letters published as a Supplementary item. When you submit your final files please clearly state in your cover letter whether or not you would like to participate in this initiative. Please note that failure to state your preference will result in delays in accepting your manuscript for publication.

Cover suggestions

COVER ARTWORK: We welcome submissions of artwork for consideration for our cover. For more

information, please see our guide for cover artwork.

Nature Cell Biology has now transitioned to a unified Rights Collection system which will allow our Author Services team to quickly and easily collect the rights and permissions required to publish your work. Approximately 10 days after your paper is formally accepted, you will receive an email in providing you with a link to complete the grant of rights. If your paper is eligible for Open Access, our Author Services team will also be in touch regarding any additional information that may be required to arrange payment for your article.

Please note that *Nature Cell Biology* is a Transformative Journal (TJ). Authors may publish their research with us through the traditional subscription access route or make their paper immediately open access through payment of an article-processing charge (APC). Authors will not be required to make a final decision about access to their article until it has been accepted. Find out more about Transformative Journals

Please use the following link for uploading these materials:
[Redacted]

Best regards,

Kendra Donahue
Staff
Nature Cell Biology

On behalf of

Melina Casadio, PhD
Senior Editor, Nature Cell Biology
ORCID ID: <https://orcid.org/0000-0003-2389-2243>

Reviewer #1:

Remarks to the Author:

The authors have performed sufficient experiments to address my concerns.

Reviewer #2:

Remarks to the Author:

In this revised submission, the authors have largely addressed my major concerns.

In particular, the evidence that Pho81 can partially compensate for Atg13/Atg17 in pexophagy is convincing. The authors also explore how phosphorylation governs the function of the putative Pho81-AKC complex through phosphoproteomics under conditions where WT or a mutant Pho81 is present. They discover three phosphorites on Atg11 are dependent upon WT Pho81 (thought to be the form engaging the AKC) and demonstrate these sites are necessary to support efficient pexophagy.

The new data exploring the consequences of the SPX domain demonstrates a clear role for this motif in intracellular localization including targeting to Atg1 puncta. Experiments intended to test a role for IP6 in regulating this interaction are consistent with the apparent negative regulation observed in imaging, though these experiments could be more fleshed out.

In total, this is a significant improvement on the already solid initial submission and will be of broad interest to the field.

Reviewer #3:

Remarks to the Author:

My comments have been adequately addressed in this revision.

I would suggest the explicit addition to the Fig 1 legend or Materials and methods that they measure the relative ratio of free GFP to total GFP

Author Rebuttal, first revision:

Reviewer #1 (Remarks to the Author):

The authors have performed sufficient experiments to address my concerns.

We thank the reviewer for the positive assessment of our work.

Reviewer #2 (Remarks to the Author):

In this revised submission, the authors have largely addressed my major concerns.

In particular, the evidence that Pho81 can partially compensate for Atg13/Atg17 in pexophagy is convincing. The authors also explore how phosphorylation governs the function of the putative

Pho81-AKC complex through phosphoproteomics under conditions where WT or a mutant Pho81 is present. They discover three phosphorites on Atg11 are dependent upon WT Pho81 (thought to be the form engaging the AKC) and demonstrate these sites are necessary to support efficient pexophagy.

The new data exploring the consequences of the SPX domain demonstrates a clear role for this motif in intracellular localization including targeting to Atg1 puncta. Experiments intended to test a role for IP6 in regulating this interaction are consistent with the apparent negative regulation observed in imaging, though these experiments could be more fleshed out.

In total, this is a significant improvement on the already solid initial submission and will be of broad interest to the field.

We thank the reviewer for the positive assessment of our work.

Reviewer #3 (Remarks to the Author):

My comments have been adequately addressed in this revision.

I would suggest the explicit addition to the Fig 1 legend or Materials and methods that they measure the relative ratio of free GFP to total GFP

We added a description of the quantifications for the GFP-base autophagy flux assays to materials and methods section.

Final Decision Letter:

Dear Dr Graef,

I am pleased to inform you that your manuscript, "A metabolite sensor subunit of the Atg1/ULK complex regulates selective autophagy", has now been accepted for publication in Nature Cell Biology.

Thank you for sending us the final manuscript files to be processed for print and online production,

and for returning the manuscript checklists and other forms. Your manuscript will now be passed to our production team who will be in contact with you if there are any questions with the production quality of supplied figures and text.

Please note that *Nature Cell Biology* is a Transformative Journal (TJ). Authors may publish their research with us through the traditional subscription access route or make their paper immediately open access through payment of an article-processing charge (APC). Authors will not be required to make a final decision about access to their article until it has been accepted. Find out more about Transformative Journals

Authors may need to take specific actions to achieve compliance with funder and institutional open access mandates. If your research is supported by a funder that requires immediate open access (e.g. according to Plan S principles) then you should select the gold OA route,

and we will direct you to the compliant route where possible. For authors selecting the subscription publication route, the journal's standard licensing terms will need to be accepted, including self-archiving policies. Those licensing terms will supersede any other terms that the author or any third party may assert apply to any version of the manuscript.

If you have not already done so, we strongly recommend that you upload the step-by-step protocols used in this manuscript to the Protocol Exchange (www.nature.com/protocolexchange), an open online resource established by Nature Protocols that allows researchers to share their detailed experimental know-how. All uploaded protocols are made freely available, assigned DOIs for ease of citation and are fully searchable through nature.com. Protocols and Nature Portfolio journal papers in which they are used can be linked to one another, and this link is clearly and prominently visible in the online versions of both papers. Authors who performed the specific experiments can act as primary authors for the Protocol as they will be best placed to share the methodology details, but the Corresponding Author of the present research paper should be included as one of the authors. By uploading your Protocols to Protocol Exchange, you are enabling researchers to more readily reproduce or adapt the methodology you use, as well as increasing the visibility of your protocols and papers. You can also establish a dedicated page to collect your lab Protocols. Further information can be found at www.nature.com/protocolexchange/about

With kind regards,

Melina Casadio, PhD
Senior Editor, Nature Cell Biology
ORCID ID: <https://orcid.org/0000-0003-2389-2243>
